# GENERATE RATHER THAN RETRIEVE: LARGE LANGUAGE MODELS ARE STRONG CONTEXT GENERATORS

**Wenhao Yu**[1*]**, Dan Iter**[2]**, Shuohang Wang**[2]**, Yichong Xu**[2]**, Mingxuan Ju**[1]**,**
**Soumya Sanyal**[3*]**, Chenguang Zhu**[2]**, Michael Zeng**[2]**, Meng Jiang**[1]
[1]University of Notre Dame [2]Microsoft Cognitive Service Research
[3]University of Southern California
[1]wyu1@nd.edu; [2]iterdan@microsoft.com

## ABSTRACT

Knowledge-intensive tasks, such as open-domain question answering (QA), require access to a large amount of world or domain knowledge. A common approach for knowledge-intensive tasks is to employ a *retrieve-then-read* pipeline that first retrieves a handful of relevant contextual documents from an external corpus such as Wikipedia and then predicts an answer conditioned on the retrieved documents. In this paper, we present a novel perspective for solving knowledge-intensive tasks by replacing document retrievers with large language model generators. We call our method *generate-then-read* (GENREAD), which first prompts a large language model to generate contextual documents based on a given question, and then reads the generated documents to produce the final answer. Furthermore, we propose a novel clustering-based prompting method that selects distinct prompts, in order to generate diverse documents that cover different perspectives, leading to better recall over acceptable answers. We conduct extensive experiments on three different knowledge-intensive tasks, including open-domain QA, fact checking, and dialogue system. Notably, GENREAD achieves 71.6 and 54.4 exact match scores on TriviaQA and WebQ, significantly outperforming the state-of-the-art *retrieve-then-read* pipeline *DPR-FiD* by +4.0 and +3.9, without retrieving any documents from any external knowledge source. Lastly, we demonstrate the model performance can be further improved by combining retrieval and generation. Our code and generated documents can be found at https://github.com/wyu97/GenRead.

## 1 INTRODUCTION

Knowledge-intensive tasks, such as open-domain question answering (QA) and fact checking, require access to a large amount of world or domain knowledge (Petroni et al., 2021). These tasks are even challenging for humans without access to an external knowledge source such as Wikipedia. A common thread of existing methods for knowledge-intensive tasks employ a *retrieve-then-read* pipeline that first retrieves a handful of relevant contextual documents from Wikipedia and then conditions the prediction of the answer on these documents along with the question (Karpukhin et al., 2020; Lewis et al., 2020; Izacard & Grave, 2021). Nevertheless, these methods mainly suffer from three drawbacks. First, candidate documents for retrieval are chunked (e.g., 100 words) and fixed, so the retrieved documents might contain noisy information that is irrelevant to the question. Second, the representations of questions and documents are typically obtained independently in modern two-tower dense retrieval models (Karpukhin et al., 2020), leading to only shallow interactions captured between them (Khattab et al., 2021). Third, document retrieval over a large corpus requires the retriever model to first encode all candidate documents and store representations for each document. These two operations limit the parameters of dense retrievers and the size of embedding vectors, and thus cannot enjoy the world knowledge or deduction capabilities of large language models (Levine et al., 2022).

---

§ Unless otherwise specified, we use the text-davinci-002 version of InstructGPT in our experiments.
* Work done during internship at Microsoft Cognitive Service Research group.

In this paper, we propose to leverage large language models, such as InstructGPT (Ouyang et al., 2022), to directly generate contextual documents for a given question, instead of retrieving relevant documents from an external corpus, such as Wikipedia. Our approach has two main advantages. First, we show that generated contextual documents contain the correct answer more often than the top retrieved documents. We believe this is because large language models generate contextual documents by performing deep token-level cross-attention between all the question and document contents, resulting in generated documents that are more specific to the question than retrieved documents. Second, we show that our approach significantly outperforms directly generating answers from large language models despite not incorporating any new external information. This is mainly because the task of generating document-level contexts is close to the objective of causal language modeling pre-training, so the world knowledge stored in the model parameters can be better utilized.

We show, on multiple datasets, that generated documents are more likely to contain correct answers than the top retrieved documents. Notably, in dense retrieval methods, as more documents are retrieved, the recall of documents containing the correct answer increases (Karpukhin et al., 2020). However, the recall performance does not scale as well with generated documents because even with sampling methods, generated documents tend to contain duplicate information. In order to improve the recall performance of generated documents, we propose a novel clustering-based prompt method. We synthesize a prompt with in-context demonstrations of question-document pairs sampled from diverse clusters. These prompts result in generated documents that cover different perspectives of the question and improve the scaling of performance as more documents are generated per question.

In contrast to the *retrieve-then-read* pipeline, our method is essentially a *generate-then-read* pipeline. Specifically, it first prompts a large language model to generate contextual documents based on a given question, and then reads the generated document to produce the final answer. The reader can still be a large model (e.g., InstructGPT (Ouyang et al., 2022)) used under a zero-shot setting, or a small one (e.g., FiD (Izacard & Grave, 2021)) fine-tuned with generated documents on the training split of the target dataset. We evaluate our proposed method on three different knowledge-intensive tasks and demonstrate its effectiveness on both zero-shot and supervised settings.

Overall, our main contributions can be summarized as follows:

1. We propose a novel *generate-then-read* pipeline for solving knowledge-intensive tasks, i.e., replacing the process of retrieving documents from Wikipedia or searching for related documents on Google, by prompting a large language model to generate relevant contextual documents.

2. We propose a novel clustering-based prompting approach to generate multiple diverse contextual documents that increases the likelihood of covering the correct answer. We demonstrate this approach can significantly improve performance on end QA and other downstream tasks.

3. We conduct extensive experiments with three knowledge-intensive NLP tasks under both zero-shot and supervised settings. Notably, our method can match or even outperform *retrieve-then-read* pipeline methods, without retrieving any documents from any external knowledge source.

## 2 RELATED WORK

KNOWLEDGE-INTENSIVE NLP VIA RETRIEVE-THEN-READ PIPELINE. Mainstream methods for solving knowledge-intensive NLP tasks employ a *retrieve-then-read* model pipeline. Given a question, this model first leverages a retriever over a large evidence corpus (e.g. Wikipedia) to fetch a set of relevant documents that may contain the answer. A reader is then used to peruse the retrieved documents and predict an answer. Recent follow-up work has mainly focused on improving the retriever (Karpukhin et al., 2020; Qu et al., 2021; Sachan et al., 2022) or the reader (Izacard & Grave, 2021; Cheng et al., 2021; Yu et al., 2022), or training the system end-to-end (Lewis et al., 2020; Singh et al., 2021). Early retrieval methods mainly employed sparse retrievers, such as BM25 (Chen et al., 2017). Recently, ORQA (Lee et al., 2019) and DPR (Karpukhin et al., 2020) have revolutionized the field by utilizing dense contextualized vectors for document indexing, leading to superior performance to traditional approaches. We propose an alternative approach which forgoes retrieval, instead extracting the knowledge from the model parameters of a large language model. We show that our approach is can be combine with dense retrievers to outperform both methods independently. Our method can also be combined with any reader mechanism, allowing generated context documents to be plugged into any current knowledge-intensive NLP pipelines.

GENERATOR AS RETRIEVER FOR OBTAINING CONTEXTUAL DOCUMENTS. Recent works have investigated using auto-regressive language models to generate identifier strings for documents, as an

intermediate target for retrievals, such as entity names (De Cao et al., 2020) or distinctive n-grams that can be mapped to full passages (Bevilacqua et al., 2022). However, one needs to create the identifiers, hence the structure was not thoroughly evaluated on a large-scale benchmark (Bevilacqua et al., 2022). Other works have demonstrated that the knowledge stored in the parameters of pre-trained language models could be "retrieved" to some extent by directly generating text (Petroni et al., 2019; Roberts et al., 2020). However, the previous work only used generation for query expansion (Mao et al., 2021), which did not exploit the potential of directly generating contextual documents for open-domain questions. Different from the above approaches that aimed to train a generator model to produce contextual document identifiers (which is still using the original Wikipedia text) or provide data augmentation to retrievers, our work directly generates contextual documents for given questions.

NLP MODELS ENHANCED BY LARGE LANGUAGE MODEL OUTPUTS. A line of recent work has shown that relevant knowledge can be elicited from large language models, especially for those domains that lack appropriate knowledge bases with sufficient coverage (Liu et al., 2022b; Fang et al., 2022). For example, Liu et al. (2022b) proposed leveraging GPT-3 to generate relevant contexts, then providing the contexts as additional input when answering a commonsense question. Another line of work focused on prompting a large language model to generate a series of intermediate reasoning steps, often referred to as *chain-of-thought* (Wei et al., 2022b; Kojima et al., 2022; Li et al., 2022). The prompt consists of an instruction (e.g., Let's think step by step!), a few demonstrations that are fixed for each task, and a new-question placeholder. The demonstrations are human-written, and each consists of a question in the style of the task and a series of intermediate reasoning steps that is helpful for answering the question. Our work does not require any human annotation, but adds to this line of work of leveraging model generated text to guide further generations. In our case, we apply this approach to knowledge-intensive tasks, which have not been explored by previous work.

## 3 PROPOSED METHOD

In this section, we present details of our proposed novel *generate-then-read* (GENREAD) pipeline for solving various knowledge-intensive tasks. Specifically, it first prompts a large language model to generate contextual documents with respect to a given query, then reads the generated documents to predict the final answer. The reader can either be a large model (e.g., InstructGPT) used for the zero-shot setting, or a small one (e.g., FiD) fine-tuned with generated documents on the training split of the target dataset. We introduce the zero-shot setting in §3.1 and supervised setting in §3.2.

### 3.1 ZERO-SHOT SETTING

Under the zero-shot setting, there is no training data – neither questions nor contextual documents. When tested on the open-domain QA task, most existing large language models directly encode the given question and predict the answer (Brown et al., 2020; Du et al., 2022; Chowdhery et al., 2022). Specifically, the question $q$, associated with some text prompt, is input to the model, which then generates the answer, denoted as $p(a|q, \theta)$, where $\theta$ represents the pre-trained model parameters. In practice, the maximum a posteriori estimation (MAP) is the final answer, i.e., $\hat{a} = \arg\max_a p(a|q, \theta)$. However, this way of directly asking large language models to output answers often leads to poor performance, as it leaves a considerable amount of additional world knowledge unexploited (Levine et al., 2022). On the contrary, the zero-shot *retrieve-then-read* pipeline first uses an off-the-shelf retriever to fetch relevant documents from an external knowledge source such as Wikipedia, then asks the large language model to read the documents and predict the answer.

In this work, we improve the performance by introducing an additional auxiliary *generated document* variable $d$, and then extend the model to have the form $p(a|q) = \sum_i p(a|d_i, q)p(d_i|q)$. In practice, we cannot sum over all possible documents $d$. Therefore, the most common approach is to compute the MAP estimate $\hat{d} = \arg\max \hat{p}(d)$ using beam search, and then to approximate the sum over $d$ with this single value. This two step approach, we label it as a *generate-then-read* pipeline.

STEP1: GENERATE. In this step, we first prompt a large language model (e.g., InstructGPT (Ouyang et al., 2022)) to generate documents based on the given question. For example, the input to the language model could be "Generate a background document to answer the given question. {question placeholder}". We can use any decoding strategy (e.g., greedy decoding, beam search), but we used greedy decoding throughout the zero-shot experiments for simplicity and reproducibility.

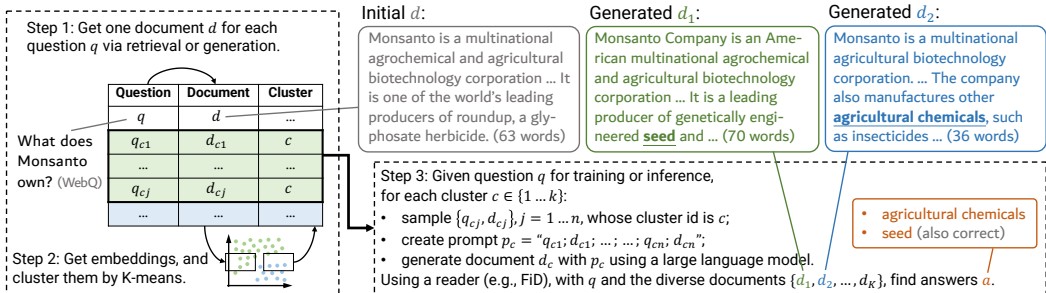

Figure 1: An overall framework of clustering-based prompting method. It leverages distinct question-document pairs sampled from each embedding cluster as in-context demonstrations to prompt a large language model to generate diverse documents, then read the documents to predict an answer.

STEP 2: READ. In the second step, we use generated sentence $\hat{d}$ along with the input question to produce the final answer from the large language model. This is actually the same setting as "zero-shot" reading comprehension, as widely studied in existing works (Brown et al., 2020; Lazaridou et al., 2022). We choose appropriate prompts from P3 (Bach et al., 2022), such as "Refer to the passage below and answer the following question. Passage: {background placeholder} Question: {question placeholder}". Finally, the language model is fed the prompted text to generate an answer.

## 3.2 SUPERVISED SETTING

Although large language models demonstrate impressive performance on zero-shot learning abilities, their performance still lag behind the supervised setting. Therefore, we also explore how the generated documents from large language models can benefit the supervised setting. As directly fine-tuning large language models on downstream datasets could be prohibitively expensive, we leverage a small reader model such as FiD to peruse the generated documents under the supervised setting.

Under the supervised setting, scaling the size of retrieved documents can lead to better performance (Karpukhin et al., 2020; Izacard & Grave, 2021). This is mainly because retrieving more documents can cover more relevant information and knowledge, i.e., a higher recall score. Nevertheless, asking a large language model to generate multiple high-quality contextual documents is a challenging task. Dense retrieval methods can fetch multiple documents covering different perspectives of the answer. Compared to dense retrievers, simply prompting a large language model to generate multiple contextual documents often leads to low knowledge coverage, since the contents generated by multiple decoding passes from the same input tend to be similar. Sampling decoding methods, such as nucleus sampling[1] (Holtzman et al., 2020) can diversify the generation process to some extent, but the knowledge content of generated texts still tends to be highly repetitive when used to generate documents for a given question. We further propose two novel solutions, including diverse human prompts and clustering-based prompts, which will be elaborated on in this section.

### 3.2.1 DIVERSE HUMAN PROMPTS

In order to avoid similar token distributions under a single prompt, we ask human annotators to provide different prompts, in order to make the generated document diverse. This method is simple, but can effectively vary the token distribution during generation. In the experiments, we empirically found this method can bring improvement to the retrieval performance (Figure 2). However, this method suffers from two drawbacks. On one hand, it requires human annotators to write different prompts, which cannot be easily generalized to different knowledge-intensive tasks. On the other hand, different large language models might be sensitive to different prompt words, which might cause a set of good prompt words not work on a different large language model.

### 3.2.2 CLUSTERING-BASED PROMPTS

To increase knowledge coverage in generated documents, we propose a novel clustering-based prompt method. It first clusters the representations of a set of documents into $K$ classes ($K = 2$ in Figure

---

[1]We treated nucleus sampling as a baseline to generate multiple documents, in which we set $p = .95$.

| Models | Open-domain QA | | | Fact Checking | | Dialogue System |
| | NQ | TriviaQA | WebQ | FEVER | FM2 | WoW (F1 / R-L) |
|---|---|---|---|---|---|---|
| *with retriever, AND directly trained on these datasets* | | | | | | | |
| DPR + InstructGPT* | 29.1 | 53.8 | 20.2 | 79.8 | 65.9 | 15.4     13.7 |
| *with retriever, BUT NOT trained on these datasets* | | | | | | | |
| BM25 + InstructGPT | 19.7 | 52.2 | 15.8 | 78.7 | 65.2 | _15.7_     13.7 |
| Contriever + InstructGPT | 18.0 | 51.3 | 16.6 | 80.4 | **66.6** | 15.5     _14.0_ |
| Google + InstructGPT | **28.8** | _58.8_ | _20.4_ | **82.9** | 66.0 | 14.8     13.2 |
| *without retriever, and not using external documents* | | | | | | | |
| Previous SoTA methods | 24.7[1] | 56.7[2] | 19.0[1] | - | - | -     - |
| InstructGPT (no docs.) | 20.9 | 57.5 | 18.6 | 77.6 | 59.4 | 15.4     13.8 |
| GENREAD (InstructGPT) | _28.0_ | **59.0** | **24.6** | _80.4_ | 65.5 | **15.8**     **14.2** |

Table 1: Zero-shot open-domain QA performance. Our proposed GENREAD with the InstructGPT reader (named GENREAD (InstructGPT)) can significantly outperform the original InstructGPT, achieving new state-of-the-art performance on three open-domain QA benchmarks (previous SoTA: [1]GLaM (Du et al., 2022), [2]FLAN (Wei et al., 2021)) under this setting without using any external document. Our GENREAD can achieve comparable or even better performance than zero-shot *retrieve-then-read* models that use a retriever or search engine to first obtain contextual documents. To ensure reproducibility, we use greedy search in decoding. All prompts used are shown in the §B.1.

1), where the number of classes is equal to the number of documents that need to be generated in the end. Next, it randomly selects $n$ question-document pairs ($n = 5$ in Figure 1) from each cluster. Lastly, a large language model presents the different $n$ question-document pairs as in-context demonstrations for generating documents to a given question. In this way, large language models are based on different distributions of examples, hence resulting in generated documents covering different perspectives. We show this in Figure 1 and illustrate the details of each step as follows.

STEP 1: GET ONE INITIAL DOCUMENT PER QUESTION. Similar to the zero-shot setting, we first ask a large language model to generate one contextual document $d$ for each question $q \in \mathcal{Q}$, where $\mathcal{Q}$ is the set of questions in the training split. Alternatively, we can use an unsupervised retriever (e.g., BM25) to obtain a document from Wikipedia. We now have a question-document pair set $\{q_i, d_i\}_{i=1}^{|\mathcal{Q}|}$.

STEP 2: ENCODE EACH DOCUMENT, DO K-MEANS CLUSTERING. We then use a large language model (i.e., GPT-3) to encode each question-document pair, i.e., $\mathbf{e}_i = \text{GPT-3}([q_i, d_i])$, resulting in a 12,288-dimensional vector per document. Then, we use K-means to cluster all embedding vectors $\{\mathbf{e}_i\}_{i=1}^{|\mathcal{Q}|}$ into $K$ sets, so each question-document pair is assigned a unique cluster id $c \in \{1, ..., K\}$. We vary the number of $K$ in the experiments, which will be illustrated in Figure 2.

STEP 3: SAMPLE AND GENERATE $K$ DOCUMENTS. Lastly, we sample $n$ question-document pairs from each cluster $c$, denoted as $\{q_{c1}, d_{c1}; q_{c2}, d_{c2}; ...; q_{cn}, d_{cn}\}$, in which $n$ is a hyperparameter[2]. Then, the $n$ sampled question-document pairs from the same cluster serve as in-context demonstrations for the large language model to generate a contextual document. For example, the input to the large language model could be "{$q_{c1}$ placeholder} {$d_{c1}$ placeholder} ... {$q_{cn}$ placeholder} {$d_{cn}$ placeholder} {input question placeholder}". By enumerating the sampled documents in these $K$ clusters, we can finally get $K$-generated documents. By conditioning on different sampled in-context demonstrations collected from different clusters, the large language model has been biased for different perspectives. Although these different perspectives exist in a latent manner, we empirically show it works well in practice, by comparing it with sampling methods, diverse human prompts (Figure 2 and Table 2) and randomly sampling $n$ pairs from the entire dataset (Table 11).

## 4 EXPERIMENTS

In this section, we conduct comprehensive experiments on three knowledge-intensive NLP tasks, including open-domain QA (NQ (Kwiatkowski et al., 2019), TriviaQA (Joshi et al., 2017) and WebQ (Berant et al., 2013)), fact checking (FEVER (Thorne et al., 2018) and FM2 (Eisenschlos et al., 2021)) and open-domain dialogue system (WoW (Dinan et al., 2019)). More detailed dataset

---

[2]In the experiments, we set $n = 5$ and found increasing $n$ does not bring extra improvement.

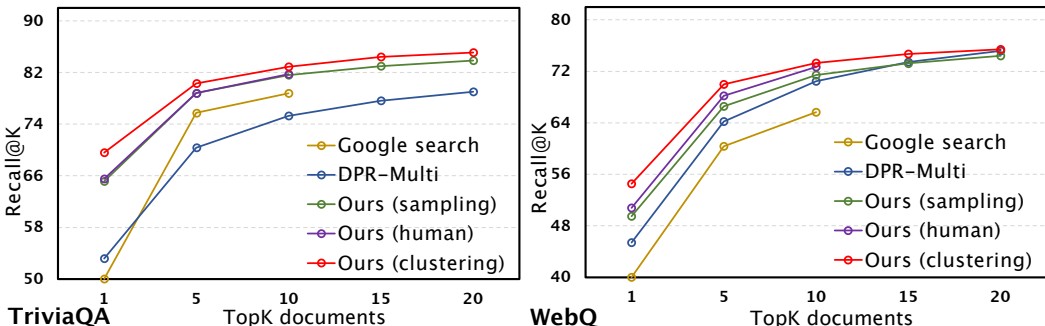

Figure 2: Recall@K on test sets, measured as the percentage of top-K documents that contain the answer. Our proposed clustering-based prompting method can outperform DPR and Google search, also two variants of using LLMs to generate documents. Exact numbers are reported in Table 5.

information can be found in Appendix A.1. To evaluate the model performance, we use exact match (EM) score for evaluating open-domain QA (Zhu et al., 2021). An answer is considered correct if and only if its normalized form has a match in the acceptable answer list. We also employ Recall@K (R@K) as an intermediate evaluation metric, measured as the percentage of top-K retrieved or generated documents that contain the answer. This metric is commonly used in evaluations of previous works (Karpukhin et al., 2020; Izacard & Grave, 2020; Sachan et al., 2022). For other knowledge-intensive tasks, we follow the KILT benchmark (Petroni et al., 2021) to use accuracy (ACC) for fact checking and F1 / Rouge-L (R-L) score for open-domain dialogue system.

## 4.1 ZERO-SHOT SETTING EXPERIMENTS

We first compare our proposed GENREAD approach with various large language models proposed in recent years, including GPT-3 (Brown et al., 2020), Gopher (Rae et al., 2021), FLAN (Wei et al., 2021), GLaM (Du et al., 2022), Chinchilla (Hoffmann et al., 2022), PaLM (Chowdhery et al., 2022) and InstructGPT (Ouyang et al., 2022). Due to the space limitation, we only put the best performance on each dataset in Table 1, in which the line is called *previous SoTA methods*. In addition, their corresponding model parameters and performance are listed in Table 9 in Appendix. All of these baseline methods use the same input formats, i.e., [prompt words; question].

GENREAD is based on InstructGPT with 175B parameters. In order to fully evaluate the effectiveness of our proposed method, we also compare with InstructGPT augmented with retrieved documents from Wikipedia or Google search. The baseline methods (1) BM25 / Contriever + InstructGPT; (2) Google + InstructGPT; (3) DPR + InstructGPT have the same input format as our GENREAD , i.e., [prompt words; contextual document; question]. BM25 is a traditional sparse retrieval method. Contriever (Izacard et al., 2022a) is a state-of-the-art unsupervised dense retrieval model. DPR (Karpukhin et al., 2020) is a supervised dense retrieval model directly trained on NQ, TriviaQA and WebQ datasets. We note that comparing with above three methods is challenging because our method only relies on the large language model itself, without using any external corpus.

### 4.1.1 EXPERIMENTAL RESULTS

In the experiments, we use InstructGPT as our backbone model. As shown in Table 1, compared with state-of-the-art large language models, our proposed GENREAD with the InstructGPT reader improves its performance by generating contextual documents and conditioning on the generated documents, even though no new data is introduced, and the generator and reader have the exact same parameters. Specifically, GENREAD can improve the EM score by +6.9 on three open-domain QA benchmarks, compared to the original InstructGPT. We also make a similar observation on fact checking and open-domain dialogue system. Our proposed GENREAD can consistently outperform the baseline InstructGPT model without retrieving any contextual documents.

To further validate the effectiveness of GENREAD , we compare against zero-shot *retrieve-then-read* pipeline models, which first use a retrieval model or the Google search engine to get a relevant contextual document, then use InstructGPT to read the texts and produce the final answer. As shown in Table 1, GENREAD can achieve on-par performance with zero-shot *retrieve-then-read* pipeline models on the NQ and FM2 datasets, and outperform them on all other benchmarks. The knowledge

| Models | # reader parameters | # docu- ments | TriviaQA open test | WebQ open test | NQ open test | Avg. |
|---|---|---|---|---|---|---|
| *baselines with retrieving from Wikipedia; all numbers reported by existing papers* | | | | | | |
| DPR (Karpukhin et al., 2020) | 110M | 100 | 56.8 | 41.1 | 41.5 | 46.5 |
| RAG (Lewis et al., 2020) | 400M | 10 | 56.1 | 45.2 | 44.5 | 48.6 |
| FiD (Izacard & Grave, 2021) | 770M | 100 | 67.6 | 50.5 | 51.4 | 56.5 |
| *baselines with retrieving from Wikipedia or Google; all numbers from our experiments* | | | | | | |
| FiD-l (DPR, Wikipedia) | 770M | 10 | 61.9 | 48.1 | 46.7 | 52.2 |
| FiD-xl (DPR, Wikipedia) | 3B | 10 | 66.3 | 50.8 | 50.1 | 55.7 |
| FiD-xl (Google search) | 3B | 10 | 70.1 | 53.6 | 45.0 | 56.2 |
| *our proposed method by leveraging a large language model to generate documents* | | | | | | |
| GENREAD (FiD-l) (sampling) | 770M | 10 | 67.8 | 51.5 | 40.3 | 53.2 |
| GENREAD (FiD-l) (clustering) | 770M | 10 | 70.2 | 53.3 | 43.5 | 55.6 |
| GENREAD (FiD-xl) (sampling) | 3B | 10 | 69.6 | 52.6 | 42.6 | 54.9 |
| GENREAD (FiD-xl) (clustering) | 3B | 10 | 71.6 | 54.4 | 45.6 | 57.1 |
| ⊢ merge retrieved documents with generated documents | | | **74.3** | **56.2** | **54.0** | **61.5** |

Table 2: Supervised open-domain QA performance. By only using generated documents from InstructGPT, our GENREAD with FiD reader (named GENREAD (FiD)) can achieve better performance than baseline methods on TriviaQA and WebQ. Through our detailed analysis of NQ, we found the performance gap mainly due to the temporality issue, which will be elaborated in §A.8.

learned by the large language models can be retrieved via autoregressive text generation. Without seeing any examples from these datasets, GENREAD can outperform using the supervised retrieval model (i.e., DPR) to recover relevant contextual documents.

## 4.2 SUPERVISED SETTING EXPERIMENTS

We compare our proposed GENREAD with *retrieve-then-read* models, including DPR (Karpukhin et al., 2020), RAG (Lewis et al., 2020), and FiD (Izacard & Grave, 2021). In addition, we compared with obtaining relevant documents from the internet using the Google search engine.

### 4.2.1 EXPERIMENTAL SETUP

For our proposed method, we replace the retriever with a large language model to directly generate contextual documents. In the experiments, we use InstructGPT (Ouyang et al., 2022). After contextual documents are retrieved or generated, we employ a FiD reader with 770M parameter models (i.e., FiD-l) and 3B parameter models (i.e., FiD-xl) that are fine-tuned on the training split of target datasets. We note that we only use 10 documents during reading for the following reasons.

*Why do we choose to use only 10 documents instead of 100 when reading?*

As noted in Section 6.2 in DPR (Karpukhin et al., 2020) and Figure 3 in FiD (Izacard & Grave, 2021), increasing the number of documents can lead to better model performance and achieve state-of-the-art when using 100 documents. However, there are two major drawbacks to using 100 documents during the reading step. First, the operation is very expensive, leading to a significant increase in memory consumption and training time. As reported by Izacard & Grave (2021), the training process requires 64 Tesla V100 32GB running for around one day. Second, generating documents by using a large language model is slow and expensive, so only using 10 documents can be a significant cost saving in our method. Therefore, in our experiments, we choose to use 10 documents during the reading process. When using FiD-770M (i.e., FiD-large), the training process can be easily performed even on a single Tesla V100 32GB GPU. Meanwhile, when only using 10 documents, we can also increase the size of FiD model from 770M to 3B, which takes about the same amount of GPU memory as using 100 documents on a 770M model, but at the same time significantly shortens the training time. We note that training T5-3B model needs a bigger cluster such as 8 Tesla V100 or A100 GPUs.

### 4.2.2 EXPERIMENTAL RESULTS ON OPEN-DOMAIN QA

We first use Recall@K to compare the retrieval accuracy of different models. As shown in Figure 2, GENREAD can significantly outperform DPR and Google search for under 10 retrieved or generated

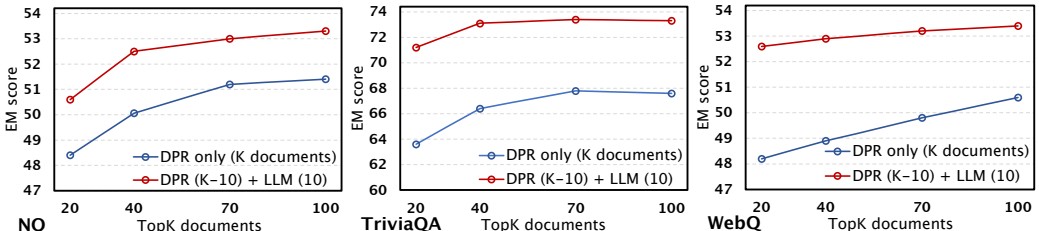

Figure 3: Combining DPR retrieved documents and large language model (LLM) generated documents can achieve significantly better performance than using DPR retrieved documents only. For a fair comparison, instead of adding LLM generated documents to the model, we replace 10 documents retrieved by DPR with 10 documents generated by LLM so the total number of documents is the same. In this experiment, we use FiD-l (i.e., FiD-large) as the reader model because when the documents scale to more than 20, FiD-xl (i.e., FiD-3B) causes out-of-memory issues on A100 GPUs.

documents. Compared to different GENREAD variants, including nucleus sampling, human written prompts, and clustering-based prompts, clustering-based prompts achieve the best performance. At the same time, we notice that the language model inevitably has the problem that the slope of the curve decreases as the number of generated documents increases. On one hand, this is due to the similarity of token distributions when large language models generate multiple documents. On the other hand, due to the shallow interaction characteristics of the dense retrieval model itself, the retrieved documents might not be completely relevant to the given question, so that the increase in recall might come from false positive documents, as also mentioned by Sachan et al. (2022).

As shown in Table 2, we can first observe the FiD model performs the best among all baseline models. Using FiD-xl with only 10 documents achieves comparable performance with using FiD-l with 100 documents. The average gap is less than 1% on three benchmarks. Compared with both close-book models and Wikipedia-based *retrieve-then-read* pipelines, our proposed GENREAD can achieve state-of-the-art performance. Furthermore, compared with using sampling methods to generate documents, the clustering-based prompt method can improve the EM score by +2.2 on average. This indicates that the clustering-based prompt method is effectively increasing the knowledge coverage of generated documents, and also leading to better downstream QA performance. We also show that GENREAD can outperform Google search on all benchmarks. We observe both our method and Google search perform worse than DPR, mainly due to the significant portion of time-dependent questions in the dataset, which is described in the following analysis.

### 4.2.3 EXPERIMENTAL RESULTS ON OTHER TASKS

We demonstrate the experimental results in Table 3. Under the supervised setting, GENREAD can achieve on par performance on the fact checking task and superior performance on the dialogue system task, indicating that large language model can be seen as a strong knowledge generator.

The main reason that GENREAD performs worse than the dense retriever

| Models | FEVER Acc. | FM2 Acc. | WoW F1 / R-L |
|---|---|---|---|
| RAG (Lewis et al., 2020) | 86.3 | 71.1 | 13.1 / 11.6 |
| FiD (Izacard & Grave, 2021) | 90.2 | 77.6 | 17.5 / 16.1 |
| GENREAD (FiD-xl) (sampling) | 89.0 | 76.3 | 18.9 / 16.7 |
| GENREAD (FiD-xl) (clustering) | 89.6 | 77.8 | 19.1 / 16.8 |
| ⊢ merge two source docs. | **91.8** | **78.9** | **20.1 / 17.9** |

Table 3: Supervised performance on fact checking (FEVER and FM2) and open-domain dialogue system (WoW).

for fact checking is that the task provides sufficient semantic information to reach strong performance on this binary decision task. So, there is a smaller semantic gap between the given factual statement and contextual documents than that of question and document pairs in open-domain QA, which is an easier retrieval setting for modern dense retrieval methods that are mainly based on vector similarity.

### 4.3 OBSERVATIONS AND EXPERIMENTAL ANALYSIS

### 4.3.1 COMPLEMENTARITY OF GENERATED AND RETRIEVED DOCUMENTS

Generated documents can be combined with retrieved documents to outperform both. Even with a very large number of retrieved documents, including few samples of generated knowledge leads to large

improvements. As shown in Table 2, merging retrieved documents with generated documents can achieve state-of-the-art performance compared to all baseline methods listed in the table. Specifically, it can improve +5.7 averagely on three open-domain QA benchmarks compared to DPR alone, and improve +4.4 averagely compared to the large language model alone.

### 4.3.2 COVERAGE ANALYSIS OVER ALL POSSIBLE ANSWERS

The improvement in open-domain QA performance is due to the fact that correct answers are included more frequently in the generated text Recall@K is the most commonly used metric in existing works to measure the retrieval performance, which computes the percentage of top-K retrieved or generated documents that contain any possible answer at least once. than in the retrieved documents. However, as many questions contain multiple correct answers, recall@K cannot fully reflect the diversity of generated or retrieved documents. Each question in the WebQ has 2.39 correct answers, 1.79 correct answers in NQ and 14.02 (including all entity alias) in the TriviaQA. NQ and WebQ do not include alias names in the labels.

In this section, we also demonstrate the answer coverage performance of different models in Table 5. Answer coverage measures the percentage of the number of answers that are contained in the documents over all possible answers. Coverage analysis showed that generated text tends to have lower coverage than retrieved documents

| Documents obtained by ↓ | NQ | TriviaQA | | WebQ |
|---|---|---|---|---|
| | - | w. alias | w/o alias | - |
| BM25 (Robertson et al., 2009) | 48.4 | 17.1 | 63.8 | 41.2 |
| Google search engine[3] | 57.9 | 18.9 | 72.0 | 54.2 |
| DPR (Karpukhin et al., 2020) | **67.9** | 17.9 | 67.3 | 58.8 |
| GENREAD (nucleus sampling) | 56.6 | 19.6 | 74.5 | 59.8 |
| GENREAD (10 human prompts) | 57.4 | 20.1 | 74.8 | 61.1 |
| GENREAD (clustering prompts) | 61.7 | **20.4** | **76.5** | **62.1** |

Table 4: Answer coverage (%) over 10 retrieved or generated documents. Case studies are provided in Tables 16-19 in Appendix.

because generated documents tends to have little diversity compared to retrieved documents. To improve coverage, we propose GENREAD with clustering, where we include examples in the prompt from different clusters of the training data to elicit more diverse generations.

## 5 EPILOGUE

CONCLUSION. In this paper, we present a novel perspective for solving knowledge-intensive tasks by replacing the dense retrieval models with large language model generators. We call it *generate-then-read*, which first prompts a large language model to generate contextual documents, then read the generated document to infer the final answer. Notably, without retrieving any documents, it reaches 71.6 and 54.4 exact match scores on TriviaQA and WebQ, significantly outperforming the current *retrieval-reader* model *DPR-FiD*, as well as on other two knowledge-intensive tasks.

LIMITATION AND FUTURE WORK. Despite the strong performance on the presented datasets, our approach is limited in its ability to update knowledge state and adapt to new domains. A major feature of *retrieve-then-read* is the ability to swap in new documents when new information is learned, such as temporally more recent documents, or adding in documents from a new domain to quickly adapt to a new downstream task. Our approach relies on a large language model to contain all this knowledge and adding new knowledge would likely require some retraining. Future work will explore how to efficiently incorporate new knowledge into our *generate-then-read* method. Besides, generated documents might suffer from hallucination error, resulting in incorrect predictions. We demonstrated case study in Table 15. Consideration in combination with recent approaches (Creswell & Shanahan, 2022) to boost generative faithfulness is a also direction worthy of future research.

## ACKNOWLEDGEMENTS

This work was supported in part by NSF IIS-2119531, IIS-2137396, IIS-2142827, CCF-1901059, and ONR N00014-22-1-2507. Wenhao is supported in part by Bloomberg Data Science Ph.D Fellowship. We are grateful to the reviewers for their insightful suggestions that have improved the quality of our paper. Their dedication to ensuring the completeness of our research is greatly appreciated.

## ETHICS STATEMENT

Large language models have a wide range of beneficial applications for society, but they also have potentially harmful applications. Previous work has shown various forms of bias, such as racial and gender bias, in large language models like GPT-3, even after explicit efforts to reduce toxic language (Chan, 2022). The importance of addressing these societal harms is acknowledged by OpenAI themselves in their 2020 paper introducing GPT-3 (Brown et al., 2020), which stated "we focus on two primary issues: the potential for deliberate misuse of language models like GPT-3 ... and issues of bias, fairness, and representation within models like GPT-3." on page 34.

The goal of this paper is to utilize knowledge stored in the parameters of large language models to answer open-domain questions and solve knowledge-intensive tasks. Unlike *retrieve-then-read* where an external corpus can be curated to be trustworthy, the use of a model to generate contextual documents may further permeate existing biases in common models. First, our work shows that generated documents suffer from challenges of stale information from outdated documents used for training. Second, we show that generated documents tend to be less diverse, potentially biasing answers towards more common entities and terms from the training data. Finally, we conducted experiments on only three large language models. It is possible that some of our conclusions or observations may not necessarily hold for other models trained with different data or objectives.

Regarding ethical solutions, future work includes (i) further exploring potential bias and intentional or unintentional harm that may result from using generated contextual documents; (ii) better aligning language models with user intent to generate less biased contents and fewer fabricated facts.

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

## A    APPENDIX

| Datasets | Splits | Train | Valid | Test | Test labels |
|---|---|---|---|---|---|
| TriviaQA (Joshi et al., 2017) | open domain | 78,785 | 8,837 | 11,313 | public |
| | wikipedia split | | | 7,993 | public |
| WebQ (Berant et al., 2013) | open domain | 3,478 | 300 | 2,032 | public |
| NQ (Kwiatkowski et al., 2019) | open domain | 79,168 | 8,757 | 3,610 | public |
| FEVER (Thorne et al., 2018) | kilt challenge | 104,966 | 10,444 | 10,100 | hidden |
| FM2 (Eisenschlos et al., 2021) | official split | 10,149 | 1169 | 1380 | public |
| WoW (Dinan et al., 2019) | kilt challenge | 63,734 | 3,054 | 2,944 | hidden |

Table 5: Datasets splits and statistics. For FEVER and WoW, labels in the test are hidden, so the model performance should be evaluated at `https://ai.facebook.com/tools/kilt/`.

### A.1    DATASETS AND SPLITS

– TRIVIAQA (TQA) (Joshi et al., 2017) contains a set of trivia questions with answers that were originally scraped from trivia and quiz-league websites.

– WEBQUESTIONS (WebQ) (Berant et al., 2013) consists of questions selected using Google Suggest API, where the answers are entities in Freebase.

– NATURAL QUESTIONS (NQ) (Kwiatkowski et al., 2019) were mined from real Google search queries and the answers are spans in Wikipedia articles identified by human annotators.

We explore the same train / dev / test splits for the open-domain QA setting as used by Izacard & Grave (2021); Karpukhin et al. (2020). For TriviaQA, GPT-3 / GLaM / PaLM (Brown et al., 2020; Du et al., 2022; Chowdhery et al., 2022) evaluate on the Wikipedia dev set of 7,993 examples, so we ran an additional evaluation on that dev set in order to compare with their performance.

– FEVER (Thorne et al., 2018) is one of the largest datasets for fact checking that requires retrieving evidence from external corpus to support if a statement is supported or refuted.

– FOOL ME TWICE (FM2) (Eisenschlos et al., 2021) is a challenging fact checking dataset collected by gamification. Players write challenging claims either entailed or refuted by evidence from Wikipedia. They are then tasked to spot the refuted claim among a group.

– WIZARD OF WIKIPEDIA (WoW) (Dinan et al., 2019) is an open-domain dialogue task for training agents that can converse knowledgeably about open-domain topics. One speaker in the conversation must ground their utterances in a specific knowledge sentence from a Wikipedia page.

We use the same train / dev / test splits in KILT challenge (Petroni et al., 2021) for the FEVER and WoW datasets. Their test labels are hidden, so the performance can only be evaluated through `https://ai.facebook.com/tools/kilt`. For FM2, we use its official dataset splits.

### A.2    IMPLEMENTATION DETAILS

We use T5-770M (Raffel et al., 2020) and T5-3B as our backbone models to implement FiD (Izacard & Grave, 2021). We use AdamW as the optimizer, with 2,000 warm-up steps. We set the dropout probability to 0.1 and weight decay to 0.01. We use one A100 for running T5-770M and set the batch size of 16. We use 8 A100 for running T5-3B and set the per GPU batch as 2, leading to the total batch size as 16. We searched different learning rates, ranging from $5e$-$6$ to $4e$-$5$, and we found $3e$-$5$ to $6e$-$5$ performed the best under the T5-3B setting and $5e$-$5$ to $1e$-$4$ performed the best under the T5-770M setting. We refer to more individual implementation details in Table 6.

We implement other baseline methods by using repositories:

– BM25: `https://github.com/castorini/pyserini`

– DPR: `https://github.com/facebookresearch/DPR`

– Contriever: `https://github.com/facebookresearch/contriever`

| Settings / Datasets | NQ | TriviaQA | WebQ | FEVER | FM2 | WoW |
|---|---|---|---|---|---|---|
| Peak learning rate | 1e-4 | 1e-4 | 1e-4 | 1e-4 | 1e-4 | 5e-5 |
| Total batch size | 64 | 64 | 64 | 64 | 64 | 16 |
| Total training steps | 15,000 | 10,000 | 10,000 | 10,000 | 10,000 | 20,000 |
| Best validation steps | 6,000 | 500 | 8,500 | 5,000 | 6,000 | 20,000 |
| Validation performance | 43.27 | 69.47 | 60.33 | 88.97 | 73.57 | 18.60 |
| Best validation ⇒ test | 43.50 | 70.22 | 53.33 | 87.25 | 74.21 | 18.49 |
| Peak learning rate | 5e-5 | 6e-5 | 3e-5 | 5e-5 | 5e-5 | 3e-5 |
| Total batch size | 16 | 16 | 16 | 16 | 16 | 8 |
| Total training steps | 20,000 | 15,000 | 15,000 | 15,000 | 15,000 | 20,000 |
| Best validation steps | 14,000 | 8,500 | 11,500 | 10,000 | 6,000 | 16,500 |
| Validation performance | 44.83 | 70.61 | 61.00 | 90.53 | 76.30 | 19.12 |
| Best validation ⇒ test | 45.55 | 71.55 | 54.36 | 89.58 | 77.78 | 18.87 |

Table 6: Hyperparaters settings and validation performance for open-domain QA (numbers reported in Table 2), fact checking and dialogue system (numbers reported in Table 3). The upper part numbers are from GENREAD (FiD-l) and the lower part numbers are from GENREAD (FiD-xl).

### A.3 REPRODUCIBILITY VIA OPEN SOURCE LARGE LANGUAGE MODELS

We note that reproducing experiments on the OpenAI API, though publicly available, costs money. For this reason, we further add an evaluation on two open-source large language models OPT (Zhang et al., 2022) and Codex (OpenAI, 2022). As shown in Table 7, OPT performed worse than InstructGPT, but still achieved comparable performance with DPR; OpenAI Codex achieved the best performance on both TriviaQA and WebQ.

| Documents obtained by ↓ | TriviaQA | WebQ |
|---|---|---|
| DPR (Karpukhin et al., 2020) | 66.3 | 50.8 |
| OPT (Zhang et al., 2022) | 62.1 | 51.8 |
| InstructGPT (Ouyang et al., 2022) | 71.3 | 54.5 |
| Codex (OpenAI, 2022) | **72.6** | **55.4** |

Table 7: Exact match (EM) score with using DPR and different open-source large language models such as OPT and Codex to generate contextual documents.

### A.4 SCALING WITH NUMBER OF LARGE LANGUAGE MODEL PARAMETERS

Figure 4 shows the scaling of performance with InstructGPT generator parameters, including Ada-150M, Babbage-1.3B, Curie-6.7B and Davinci-175B. We note that for both FiD and our GENREAD , we use the FiD-xl with 10 input documents either retrieved from Wikipedia or generated by InstructGPT. The performance of both TriviaQA and WebQ continues to improve as the generator model parameters increase, as does the slope. Only with the largest size InstructGPT, GENREAD can outperform the *DPR-FiD*. This indicates using large language model to generate contextual documents is an "emergent ability" of scaling, which is not present in smaller models but is only present in larger language models (Wei et al., 2022a).

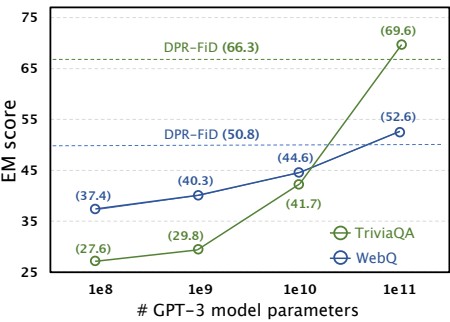

Figure 4: Model performance with different size of InstructGPT as context generators.

## A.5 Readability Analysis of Retrieved and Generated Documents

After we manually compare some retrieved documents from DPR and generated documents from InstructGPT, we observe that the readability of different documents, when they contain the correct answer string, is different. In other words, documents containing answers might also contain noisy information that is irrelevant to the question, which could affect both the model and human reading.

| Documents obtained by ↓ | NQ | TriviaQA | WebQ |
|---|---|---|---|
| DPR (Karpukhin et al., 2020) | 63.1 | 80.2 | 63.3 |
| GENREAD (nucleus sampling) | 58.7 | 83.7 | 63.8 |
| GENREAD (clustering prompts) | **64.0** | **86.8** | **66.7** |

Table 8: Readability study on retrieved documents and generated documents. See detailed analysis in §A.5.

In order to further validate the readability of retrieved documents and generated documents, we extracted a subset of data examples from NQ, TriviaQA and WebQ datasets, in which both retrieved and generated documents contain the correct answer. As shown in Table 8, when both retrieved and generated documents contain the correct answer, the FiD reader can produce more correct answers when reading the generated documents from large language models (e.g., InstructGPT).

We also provide some case studies in Tables 16-19. For example, in Table 18, the question is "What city was Zeus the patron god of?". The first document retrieved by DPR is "Like the other Panhellenic Games, the ancient Olympic Games were a religious festival, held at the sanctuary of Zeus at Olympia.". Although it contains the correct answer, it is hard to infer the answer "Olympia" from it. On the contrary, InstructGPT generates the document "Zeus was the patron god of the city of Olympia, which was located in the northwestern Peloponnese region of Greece. Olympia was the site of the Olympic Games, held every four years in honor of Zeus.", which is much easier to read.

## A.6 Additional Numbers for Tables in the Main Paper

– Table 9 contains additional evaluation results for Table 1. It demonstrates zero-shot open-domain QA performance, compared to recent large language model.

– Figure 5 contains additional retrieval performance evaluation for Figure 3 of experiments on combining DPR retrieved documents and large language model generated document.

– Table 10 contains additional retrieval performance evaluated by Recall@K of baselines and different GENREAD variants. Some numbers in the table overlaps with those in Figure 2.

| Models | # total parameters | NQ open test | TriviaQA open test | TriviaQA wiki split | WebQ open test |
|---|---|---|---|---|---|
| GPT-3 (Brown et al., 2020) | 175B | 14.6 | 49.2 | 64.3 | 14.4 |
| Gopher (Rae et al., 2021) | 280B | 10.1 | 43.5 | 52.8 | - |
| FLAN (Wei et al., 2021) | 137B | 20.7 | 56.7 | 68.1 | - |
| GLaM (Du et al., 2022) | 64B | 21.5 | - | 68.0 | 19.0 |
| Chinchilla (Hoffmann et al., 2022) | 70B | 16.6 | 55.4 | 67.0 | - |
| PaLM (Chowdhery et al., 2022) | 540B | 21.2 | - | **76.9** | 10.9 |
| InstructGPT (Ouyang et al., 2022) | 175B | 19.5 | 57.4 | 68.5 | 19.9 |
| GENREAD (InstructGPT) | 175B | **28.2** | **59.3** | 70.3 | **24.8** |

Table 9: Additional numbers for Table 1. Zero-shot open-domain QA performance, compared to recent large language models. All models in the table do not leverage any external corpus for document retrieval. Compared to InstructGPT, our proposed GENREAD can improve the EM score by +6.9 on average. GENREAD can achieve state-of-the-art performance on open test sets.

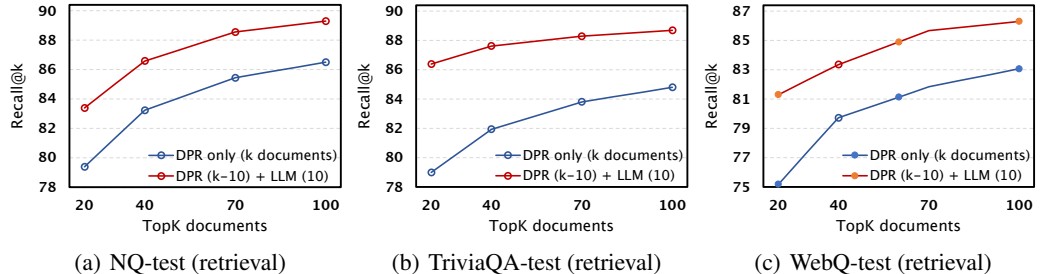

(a) NQ-test (retrieval)   (b) TriviaQA-test (retrieval)   (c) WebQ-test (retrieval)

Figure 5: Additional retrieval performance evaluation for Figure 3 of experiments on combining DPR retrieved documents and large language model generated documents. Merging documents from two sources achieved significantly better performance than using DPR retrieved documents only.

| Models | TriviaQA R@1 | R@10 | R@20 | WebQ R@1 | R@10 | R@20 | NQ R@1 | R@10 | R@20 |
|---|---|---|---|---|---|---|---|---|---|
| BM25 (Robertson et al., 2009) | 46.2 | 71.7 | 76.4 | 19.1 | 51.8 | 62.6 | 22.3 | 55.6 | 63.9 |
| Contriever (Izacard et al., 2022a) | 34.0 | 67.9 | 74.3 | 18.2 | 55.7 | 65.7 | 18.8 | 54.8 | 65.1 |
| DPR (Karpukhin et al., 2020) | 53.2 | 75.3 | 79.0 | 45.4 | 70.5 | 75.2 | 44.6 | **74.5** | **79.5** |
| Google Search engine API | 50.0 | 78.8 | - | 40.0 | 65.6 | - | 35.5 | 67.5 | - |
| GENREAD (nucleus, p=.95) | 65.1 | 81.6 | 83.8 | 49.5 | 71.4 | 74.4 | 40.1 | 66.2 | 70.6 |
| GENREAD (10 human prompts) | 65.5 | 81.8 | - | 50.8 | 72.7 | - | 40.5 | 66.9 | - |
| GENREAD (clustering prompts) | **69.6** | **82.9** | **85.1** | **54.5** | **73.3** | **75.4** | **48.0** | 70.9 | 74.5 |

Table 10: Retrieval performance evaluated by Recall@K of baselines and different GENREAD variants. Some numbers in the table overlaps with those in Figure 2. The table aims to show the performance of more methods, and to provide accurate recall numbers for future research comparisons.

| | TriviaQA R@10 | EM | WebQ R@10 | EM | NQ R@10 | EM |
|---|---|---|---|---|---|---|
| Sample 5 documents from *entire data* | 81.5 | 70.9 | 72.5 | 53.3 | 69.2 | 44.2 |
| Sample 5 documents from *each cluster* | 82.7 | 71.8 | 73.3 | 54.4 | 70.6 | 45.3 |

Table 11: Ablation study on the strategy of sampling documents as in-context demonstrations.

A.7 DISCUSSION ON INFERENCE COST OF DPR AND INSTRUCTGPT

We now compare the costs of using DPR and InstructGPT to retrieve or generate contextual documents. We consider DPR using the BERT-base (Devlin et al., 2019) version with 110M parameters and InstructGPT using its largest version with 175B parameters. For simplicity, we use the FLOPs-per-token estimates for Transformer-based language models, which is introduced by Kaplan et al. (2020). It should be noted that FLOPs are not a direct measure of real-world computing costs, as latency, power consumption, and other costs can vary widely based on other factors (Liu et al., 2022a).

For the DPR model, all Wikipedia documents (around 21M) only need to be encoded once. Therefore, as the number of input questions increases, the marginal computational cost gradually decreases. For fair comparison, we first use DPR to encode all 21M Wikipedia documents once. Encoding all Wikipedia documents requires $110e6$ (BERT-base parameters) $\times 21e6$ (total number of documents) $\times 100$ (tokens per document) $= 2.3e17$ FLOPs. When the embedding of all candidate documents are produced, retrieving documents for a given question requires $110e6$ (BERT-base parameters) $\times 20$ (tokens per question) $+21e6$ (total number of documents) $\times (768 + 768 - 1) = 3.2e10$ FLOPs.

For InstructGPT, it requires $175e9$ (InstructGPT parameters) $\times 10$ (number of documents) $\times 55$ (generated tokens per document) $= 9.6e13$ FLOPs to generate 10 documents for a given question.

Therefore, the equation for the total cost $Y_{\text{DPR-cost}}$ to retrieve 10 documents using DPR versus the number of input questions $X$ is: $Y_{\text{DPR-cost}} = 3.2e10X + 2.3e17$. Besides, the equation for the total cost $Y_{\text{GPT3-cost}}$ to generate 10 documents using InstructGPT versus the number of input questions $X$ is: $Y_{\text{GPT3-cost}} = 9.6e13X$. When $Y_{\text{DPR-cost}} = Y_{\text{GPT3-cost}}$, $X \approx 2473$. In conclusion, if the number of input questions is less than 2473, the total cost of InstructGPT is lower than the DPR; if the number of input questions is greater than 2473, the total cost of InstructGPT exceeds the DPR.

A.8 ERROR ANALYSIS AND CASE STUDIES ON THE NQ DATASET

As stated in Zhang & Choi (2021), NQ contains a significant proportion, roughly 16.5%, of questions that have time-dependent answers. Similarly, Izacard et al. (2022b) observed using the latest version of Wikipedia (12 / 2021) could lead to 4.4 drops of the EM score, compared to the Wikipedia version (12 / 2018) that the NQ questions are created from. We provide case studies in Table 13 in Appendix.

We did case studies of 100 examples from the NQ dataset. The results are shown in Table 12. Among these 100 examples, we found that 29 examples have data collection and annotation mistakes, mainly including the temporal question issue (13 / 29) and the incomplete answer issue (16 / 29). A typical temporal-dependent question is that no specific time condition is provided. For example, "Who won the MVP for the National League?" could have different answers in different years. In 2017, the MVP is Giancarlo Stanton, and in 2018, the MVP is Christian Yelich. Besides, some answer labels provided in the NQ dataset are not complete. For example, person names in the NQ dataset usually consist of first, middle, and last names, but most names in the generated documents are first and last names. For the question "who played lionel in as time goes by?", the labeled answer is "Geoffrey Dyson Palmer". *DPR-FiD* produces "Geoffrey Dyson Palmer" but GENREAD produces "Geoffrey Palmer", both of which should be considered correct. More examples are provided in Table 14.

Besides, GENREAD produced correct answers for 49 questions. Among the 22 incorrect predictions, 12 of them could be classified as retrieval errors (i.e., step-I error) and 12 as reading errors (i.e., step-II error). In all cases of retrieval errors, none of the generated documents contain the correct answer. In all cases of reading errors, at least one generated document contains the correct answer but the reader model failed to infer the correct answer from the documents..

| | | | |
|---|---|---|---|
| Good Q&A (71%) | Correct prediction (49%) | | - Query: Who got the first Nobel Prize in Physics?
- Document: The first Nobel Prize in Physics was awarded in 1901 to Wilhelm Conrad Röntgen for his discovery of the remarkable rays subsequently named after him.
- Predicted answer: Wilhelm Conrad Röntgen
- Correct answer: Wilhelm Conrad Röntgen |
| | Wrong retrieval (12%) | Hallucinations (8%) | - Query: Who died in the first episode of Stranger Things?
- Document: In the first episode of Stranger Things, the character Will Byers dies. He is killed by Demogorgon, a monster from the Upside Down.
- Predicted answer: Will Byers
- Correct answer: Benny Hammond |
| | | No hit answers (4%) | - Query: When was coffee first made into a drink?
- Document: The history of coffee goes back to the 10th century, with coffee trees native to Ethiopia. The earliest substantiated evidence of either coffee drinking or knowledge of coffee tree is from sixth century AD in Ethiopia.
- Predicted answer: the 10th century
- Correct answer: the 15th century |
| | Wrong reading (10%) | | - Query: When is the fourth movie of the Divergent series coming out?
- Document: The fourth movie in the Divergent series was originally scheduled to be released in June 2017, but was delayed indefinitely.
- Predicted answer: June 2017
- Correct answer: never made |
| Bad Q&A (29%) | Temporal questions (13%) | | - Query: Who won the MVP for the National League?
- Document: In 2017, the National League MVP was won by Giancarlo Stanton of the Miami Marlins. In 2018, the National League MVP was won by Christian Yelich of the Milwaukee Brewers.
- Predicted answer: Christian Yelich
- Correct answer: Giancarlo Stanton |
| | Incomplete answers (16%) | | - Query: Where do the greasers live in the Outsiders?
- Document: The Outsiders is a novel by S.E. Hinton. It is about a gang of greasers in Oklahoma in the 1960s. The National League MVP was won by Christian Yelich of the Milwaukee Brewers.
- Predicted answer: Oklahoma
- Correct answer: Tulsa, Oklahoma |

Table 12: Case study on 100 GENREAD predictions in the NQ dataset. Among 100 examples, there are 49 correct predictions, i.e., EM = 49%. We further categorized 51 incorrect predictions of our GENREAD, including errors caused by data collection and annotation, and errors caused by model prediction. In addition, we provide more case studies in Tables 13-15 (Table 13 for the temporal question issue; Table 14 for the incomplete answer issue; Table 15 for the hallucination issue).

| Original question | NQ labels | Correct labels |
|---|---|---|
| **Q:** When is the last time the philadelphia won the superbowl? 
 **DPR:** 2017 ✘; **Google search:** 2018 ✔; **GENREAD :** February 4, 2018 ✔ | Super Bowl LII; 2017 | 2018; February 4, 2018 |
| **Q:** Who has the most big ten championships in football? 
 **DPR:** Michigan ✘; **Google search:** Ohio State ✔; **GENREAD :** Ohio State ✔ | Michigan | Ohio State |
| **Q:** Who has the most super bowls in nfl history? 
 **DPR:** Pittsburgh Steelers ✔; **Google search:** New England Patriots ✔; **GENREAD :** New England Patriots ✔ | Pittsburgh Steelers | Pittsburgh Steelers; New England Patriots |
| **Q:** How many casinos are in atlantic city new jersey? 
 **DPR:** eleven ✘; **Google search:** nine ✔; **GENREAD :** nine ✔ | 11; eleven | 9; nine |
| **Q:** When did the us not go to the olympics? 
 **DPR:** 1980 ✔; **Google search:** 1980 and 1984 ✔; **GENREAD :** 1984 ✔ | 1980 | 1980; 1984 |
| **Q:** Largest cities in the world by population? 
 **DPR:** Beijing ✘; **Google search:** Tokyo ✔; **GENREAD :** Tokyo ✔ | Beijing | Tokyo |
| **Q:** Who has most followers on instagram in world? 
 **DPR:** Instagram ✘; **Google search:** Cristiano Ronaldo ✔; **GENREAD :** Cristiano Ronaldo ✔ | Selena Gomez | Cristiano Ronaldo |
| **Q:** Who is the no. 1 ranked tennis player in the world? 
 **DPR:** Rafael Nadal ✘; **Google search:** Novak Djokovic ✔; **GENREAD :** Novak Djokovic ✔ | Rafael Nadal | Novak Djokovic |

Table 13: Case studies of temporality issues of the NQ dataset. All these questions are drawn from Zhang & Choi (2021), which contains a subset of NQ data examples with time-dependent questions.

| Original question | *DPR-FiD* predictions | GENREAD predictions |
|---|---|---|
| **Q:** Who played lionel in as time goes by? | Geoffrey Dyson Palmer | Geoffrey Palmer |
| **Explanation:** The labeled answer is "Geoffrey Dyson Palmer", however, "Geoffrey Palmer" is also correct. 
 **DPR retrieved documents:** Geoffrey Dyson Palmer, (born 4 June 1927) is an English actor known for his roles in British television sitcoms playing Jimmy Anderson in "The Fall and Rise of Reginald Perrin", Ben Parkinson in "Butterflies" and Lionel Hardcastle in "As Time Goes By". His film appearances include "A Fish Called Wanda", "The Madness of King George", "Mrs. Brown", and "Tomorrow Never Dies". 
 **GPT generated documents:** As Time Goes By is a British sitcom that aired on BBC One from 1992 to 2005. The show starred Geoffrey Palmer and Judi Dench as Lionel and Jean Pargetter, a middle-aged couple who reunite after many years apart. Lionel was played by Palmer, who was also a writer on the show. | | |
| **Q:** How many cracker barrels in the united states? | 645 | over 630 |
| **Explanation:** The labled answer is "639" or "over 600", so "over 630" is also a reasonable answer. | | |
| **Q:** Where do the greasers live in the outsiders? | Tulsa, Oklahoma | Oklahoma |
| **Explanation:** The labled answer is "Tulsa, Oklahoma", but "Oklahoma" is also a correct answer. 
 **DPR retrieved documents:** The movie received mostly positive reviews from critics, and performed well at the box office, grossing 33 million on a 10 million budget. In Tulsa, Oklahoma, greasers are a gang of tough, low-income working-class teens. They include Ponyboy Curtis and his two older brothers, Sodapop and Darrel, as well as Johnny. 
 **GPT generated documents:** The Outsiders is a novel by S.E. Hinton. It is about a gang of greasers in Oklahoma in the 1960s. The greasers are from the poor side of town and constantly in trouble with the law. | | |
| **Q:** Where are unipolar neurons found in spinal cord? | the granule region | dorsal root ganglia |
| **Explanation:** The labled answer is "the distal dorsal root", but the output "dorsal root ganglia" is the same. | | |

Table 14: Case studies of the incomplete answers issue of the NQ dataset. Since the labels in NQ are spans identified from Wikipedia passages, it is easier for *DPR-FiD* to predict correct answers.

**Question:** Who wrote the first declaration of human rights? **Answer:** Cyrus Cylinder

**Generated document:** The first declaration of human rights was the Virginia Declaration of Rights, which was written by George Mason in 1776.

**Retrieved document:** John Peters Humphrey, OC (April 30, 1905 - 2013 March 14, 1995) was a Canadian legal scholar, jurist, and human rights advocate. He is most famous as the author of the first draft of the Universal Declaration of Human Rights. ... ... (Wikipedia page: John Peters Humphrey)

**Golden Wikipedia evidence:**

- George Mason's writings, including substantial portions of the Fairfax Resolves of 1774, the Virginia Declaration of Rights of 1776, and his Objections to this Constitution of Government (1787) opposing ratification, have exercised a significant influence on American political thought and events. Mason prepared the first draft of the Virginia Declaration of Rights in 1776, and his words formed much of the text adopted by the final Revolutionary Virginia Convention. (Wikipedia page: George Mason)

- The Cyrus Cylinder was dubbed the "first declaration of human rights" by the pre-Revolution Iranian government, a reading prominently advanced by Shah Mohammed Reza Pahlavi, in a 1967 book, The White Revolution of Iran. (Wikipedia page: Cyrus Cylinder)

**Hallucination:** In GPT generated documents, *first declaration of human rights* was considered as *Virginia Declaration of Rights*, which is factually incorrect. This error might occur because Wikipedia page of George Mason contains similar descriptions, causing the model to confuse their meanings.

**Model predictions:** *DPR-FiD* John Peters Humphrey; GENREAD George Mason

**Question:** Who died in the first episode of stranger things **Answer:** Benny *or* Barb

**Generated document:** In the first episode of Stranger Things, the character Will Byers dies. He is killed by the Demogorgon, a monster from the Upside Down.

**Retrieved document:** ... ... was the character of Barb, the nerdy friend and classmate of Nancy who is taken and killed by the monster early in the season. According to actress Shannon Purser, Barb wasn't supposed to be a big deal, and the Duffer Brothers had not gone into great detail about the character since the focus was on finding Will. ... ... (Wikipedia page: Stranger Things)

**Golden Wikipedia evidence:**

One area of focus from these fans was the character of Barb, the nerdy friend and classmate of Nancy who is taken and killed by the monster early in the season. According to actress Shannon Purser, Barb wasn't supposed to be a big deal, and the Duffer Brothers had not gone into great detail about the character since the focus was on finding Will. ... ... (Wikipedia page: Stranger Things)

The owner, Benny, learns from a tattoo of "011" on her arm that her name is Eleven. Brenner, monitoring the phone lines, sends agents to the diner after Benny calls social services. The agents kill Benny, but Eleven manages to escape using telekinetic abilities. Joyce's phone short circuits after receiving a mysterious phone call that she believes is from Will. While searching for Will in the woods, Mike, Dustin and Lucas come across Eleven. (Wikipedia page: Stranger Things)

**Hallucination:** In GPT generated documents, *Will Byers* is said to be dead, which is actually incorrect. Will Byers is actually a fake deaths in season 1 and he is still alive at the end of the season.

**Model predictions:** *DPR-FiD* Barb; GENREAD Will

**Question:** Who played the judge in good will hunting? **Answer:** James P. Flynn

**Generated document:** The character of the Judge in Good Will Hunting was played by actor Robert Sean Leonard. Leonard is best known for his role as Dr. James Wilson in the television series House.

**Retrieved document:** Janitor could show such facility with matrices applied this way. Good Will Hunting Good Will Hunting is a 1997 American drama film, directed by Gus Van Sant, and starring Robin Williams, Matt Damon, Ben Affleck, ... ... (Wikipedia page: Good Will Hunting)

**Golden Wikipedia evidence:**

- James P. Flynn appeared in many films shot in the New England area. Flynn was cast as a judge in the Boston-based film Good Will Hunting in 1997. Later, he acted in the 1999 film The Cider House Rules and What's the Worst That Could Happen? in 2001. (Wikipedia page: James P. Flynn)

- Cast: James P. Flynn as Judge George H. Malone (Wikipedia page: Good Will Hunting)

**Hallucination:** In GPT generated documents, *Robert Sean Leonard* is considered as a *judge in Good Will Hunting*, which is factually incorrect. This error leads to wrong answer prediction.

**Model predictions:** *DPR-FiD* Stellan Skarsgrd; GENREAD Alan Arkin

Table 15: Case studies of hallucination errors in InstructGPT generated documents. The documents contain contents that contradict to the facts and world knowledge, resulting in wrong predictions.

**Query:** What type of music did Mozart composed?

**Answers:** opera, classical music, art song, chamber music, ballet

**DPR #1:** The following is a list of surviving works that Mozart composed for performance at gatherings of Masons. The story and music of his opera "The Magic Flute" is also considered to have strong Masonic influences. The following is a partial list of family members, patrons, and colleagues who were Masons. Mozart's grandfather Johann Georg, a bookbinder, was raised among the extended Mozart family in Augsburg, in the house of Johann's own grandfather David Mozart. David and his children were distinguished architects and master ... (Wikipedia id: 10988168 – "Mozart and Freemasonry")

**DPR #2:** Progressively, and in large part at the hands of Mozart himself, the contrapuntal complexities of the late Baroque emerged once more, moderated and disciplined by new forms, and adapted to a new aesthetic and social milieu. Mozart was a versatile composer, and wrote in every major genre, including symphony, opera, the solo concerto, chamber music including string quartet and string quintet, and the piano sonata. These forms were not new, but Mozart advanced their technical sophistication and emotional reach. He almost single-handedly developed and popularized the Classical piano concerto. He wrote a great deal ... (Wikipedia id: 451968 – "Wolfgang Amadeus Mozart")

**Among top-10 retrieved documents:** opera: 2 times   chamber music: 1 time

**GPT nucleus #1:** Mozart was a prolific composer and wrote in many different genres. He composed symphonies, operas, concertos, chamber music, and more. Many of his works are considered some of the greatest ever written in those genres.

**GPT nucleus #2:** Mozart composed a variety of music, including operas, symphonies, chamber music, concertos, and sonatas. His early works were influenced by the music of Haydn and Bach, and his later works show the influence of the French and Italian composers of his time. Mozart's music is characterized by its beauty, elegance, and precision, and he is considered one of the greatest composers of all time.

**GPT nucleus #3:** Mozart composed a wide variety of music, including operas, symphonies, concertos, solo pieces, and chamber music. He also wrote a number of religious works, including masses and motets.

**Among top-10 generated documents:** chamber music: 9 times

**GPT cluster #1:** Mozart found work in Vienna as a piano teacher and composer for opera and other works. There he met a fellow Mason, Joseph Haydn, who was to become an important mentor. Mozart married Constanze Weber in 1782; their marriage is thought to have been happy, in contrast to his earlier unhappy arranged marriage to his cousin, Maria Anna ÑannerlM̈ozart. Mozart composed more than 600 works, including works widely.

**GPT cluster #2:** Mozart was a prolific composer, and wrote in a wide variety of genres. He composed over 600 works, including operas, symphonies, concertos, solo works, and chamber music. His operas include The Marriage of Figaro, Don Giovanni, and Cosi fan tutte. His symphonies include the well-known Symphony No. 40 in G minor, and his concertos include the Piano Concerto No. 21 in C major, known as the Ëlvira Madigan̈concerto.

**GPT cluster #3:** Mozart was a prolific composer who wrote music in a wide variety of genres. He is best known for his work in the classical genre, which includes such works as his Symphony No. 40 in G minor and his piano concertos. He also wrote operas, including The Marriage of Figaro and The Magic Flute. In addition to his work in classical music, Mozart also composed works in the popular genre, such as his Eine kleine Nachtmusik.

**Among top-10 generated documents:** opera: 2 times   chamber music: 7 times   classical music: 1 time

Table 16: Case study of retrieved documents and generated documents from language models (1/4).

**Query:** What does Monsanto own?

**Answers:** agricultural chemicals, seed, agriculture, chemical industry

**DPR #1:** In September 2016, Monsanto agreed to be acquired by Bayer for US$66 billion. In an effort to receive regulatory clearance for the deal, Bayer announced the sale of significant portions of its current agriculture businesses, including its seed and herbicide businesses, to BASF. The deal was approved by the European Union on March 21, 2018, and approved in the United States on May 29, 2018. The sale closed on June 7, 2018; Bayer announced its intent to discontinue the Monsanto name, with the combined company operating solely under the Bayer brand. ... (Wikipedia id: 935126 – "Monsanto")

**DPR #2:** Behind it all is of course an evil corporation: Monsanto. GMO conspiracy theories GMO conspiracy theories are conspiracy theories related to the production and sale of genetically modified crops and genetically modified food (also referred to as genetically modified organisms or "GMOs"). These conspiracy theories include claims that agribusinesses, especially Monsanto, have suppressed data showing that GMOs cause harm, deliberately cause food shortages to promote the use of GM food, or have co-opted government agencies ... (Wikipedia id: 19304897 – "GMO conspiracy theories")

**Among top-10 retrieved documents:** agriculture: 1 time  seed: 2 times

**GPT nucleus #1:** Monsanto is an American agricultural biotechnology corporation founded in 1901. It is headquartered in Creve Coeur, Missouri. Monsanto is a leading producer of genetically engineered seed and Roundup, a glyphosate-based herbicide.

**GPT nucleus #2:** Monsanto is a multinational agrochemical and agricultural biotechnology corporation headquartered in Creve Coeur, Missouri. It is one of the world's leading producers of Roundup, a glyphosate-based herbicide. The company also manufactures genetically engineered seed and Roundup Ready crops, including corn, soybeans, and cotton. As of 2018, Monsanto is the world's largest producer of glyphosate. In 2018, Bayer acquired Monsanto for $63 billion.

**Among top-10 generated documents:** seed: 6 times

**GPT cluster #1:** Monsanto Company is an American multinational agrochemical and agricultural biotechnology corporation headquartered in Creve Coeur, St. Louis, Missouri. It is a leading producer of genetically engineered (GE) seed and of the herbicide glyphosate, which it markets under the Roundup brand. Founded in 1901 by John Francis Queeny, Monsanto spent most of the 20th century as a chemical company, becoming one of the world's leading manufacturers of plastics, resins, and other chemicals.

**GPT cluster #2:** Monsanto is a biotechnology and agriculture company that specializes in genetically modified organisms (GMOs). The company owns a number of patents on GMOs, and it produces a range of products that contain GMOs, including herbicides, pesticides, and seeds.

**GPT cluster #3:** Monsanto is a multinational agricultural biotechnology corporation. It is one of the world's leading producers of genetically engineered seed and Roundup, a glyphosate-based herbicide. The company also manufactures other agricultural chemicals, such as insecticides and fungicides.

**Among top-10 generated documents:** agriculture: 2 times  seed: 5 times  agricultural chemicals: 1 time

Table 17: Case study of retrieved documents and generated documents from language models (2/4).

**Query:** What city was Zeus the patron god of?
**Answers:** Olympia, Elis

**DPR #1:** Many were specific only to a particular deity or city-state. For example, the festival of Lykaia was celebrated in Arcadia in Greece, which was dedicated to the pastoral god Pan. Like the other Panhellenic Games, the ancient Olympic Games were a religious festival, held at the sanctuary of Zeus at Olympia. Other festivals centred on Greek theatre, of which the Dionysia in Athens was the most important. More typical festivals featured a procession, large sacrifices and a feast to eat the offerings, and many included entertainments and customs such as visiting friends, wearing fancy dress and unusual behaviour in the ... (Wikipedia id: 1945872 – "Ancient Greek religion")

**DPR #2:** Theatre of Dionysus The Theatre of Dionysus Eleuthereus is a major theatre in Athens, considered to be the world's first theatre, built at the foot of the Athenian Acropolis. Dedicated to Dionysus, the god of plays and wine (among other things), the theatre could seat as many as 17,000 people with excellent acoustics, making it an ideal location for ancient Athens' biggest theatrical celebration, the Dionysia. It was the first theatre ever built, cut into the southern cliff face of the Acropolis, and supposedly the birthplace of Greek tragedy. The remains of a restored and redesigned Roman version can still ... (Wikipedia id: 3732515 – "Theatre of Dionysus")

**Among top-10 retrieved documents:** Olympia: 1 time

**GPT nucleus #1:** Zeus was the patron god of the city of Olympia in Greece.

**GPT nucleus #2:** The city of Olympia in Greece was the site of the ancient Olympics, and it was also the home of Zeus, the patron god of the Olympics.

**Among top-10 generated documents:** Olympia: 6 times

**GPT cluster #1:** Zeus was the patron god of the city of Olympia, which was located in the northwestern Peloponnese region of Greece. Olympia was the site of the Olympic Games, which were held every four years in honor of Zeus.

**GPT cluster #2:** Zeus was the patron god of the city of Olympia. Olympia was a sanctuary of ancient Greece in Elis on the Peloponnesos peninsula, in south-western Greece. The sanctuary was known as the Panhellenic Sanctuary of Zeus, and the altar of Zeus was located in the Temple of Zeus. The Olympic Games were held in Olympia every four years from 776 BC to 393 AD.

**GPT cluster #3:** Zeus was the patron god of the city of Olympia.

**Among top-10 generated documents:** Olympia: 9 times Elis: 1 time

Table 18: Case study of retrieved documents and generated documents from language models (3/4).

**Query:** What is the collection of the districts to the east of the Jordan River?
**Answers:** Jordan, Golan Heights

**DPR #1:** Water politics in the Jordan River basin Water politics in the Jordan River basin refers to political issues of water within the Jordan River drainage basin, including competing claims and water usage, and issues of riparian rights of surface water along transnational rivers, as well as the availability and usage of ground water. Water resources in the region are scarce, and these issues directly affect the five political subdivisions (Israel, the West Bank, Lebanon, Syria and Jordan) located within and bordering the basin, which were created since the collapse, , during World War I, of the former single controlling entity, the ... (Wikipedia id: 12822413 – "Water politics in the Jordan River basin")

**DPR #2:** Jordan's western border follows the bottom of the rift. Although an earthquake-prone region, no severe shocks had been recorded for several centuries. By far the greatest part of the East Bank is desert, displaying the land forms and other features associated with great aridity. Most of this land is part of the Syrian Desert and northern Arabian Desert. There are broad expanses of sand and dunes, particularly in the south and southeast, together with salt flats. Occasional jumbles of sandstone hills or low mountains support only meager and stunted vegetation that ... (Wikipedia id: 210052 – "Geography of Jordan")

**Among top-10 retrieved documents:** Jordan: 8 times

**GPT nucleus #1:** The collection of the districts to the east of the Jordan River is known as the West Bank. The West Bank is a landlocked territory situated in Western Asia. It is bordered by Jordan to the east, Egypt to the southwest, and Israel to the west and north.

**GPT nucleus #2:** The collection of the districts to the east of the Jordan River is the West Bank. The West Bank is a landlocked territory near the Mediterranean coast of Western Asia, bordered by Lebanon to the north, Syria to the northeast, Jordan to the east, and Egypt to the southwest. It is the smaller of two political entities that comprise the Palestinian territories, the other being the Gaza Strip.

**Among top-10 generated documents:** Jordan: 10 times

**GPT cluster #1:** The collection of the districts to the east of the Jordan River is known as the West Bank. The West Bank is a landlocked territory situated in Western Asia. It is bordered by Jordan to the east, Egypt to the southwest, and Israel to the west and north.

**GPT cluster #2:** The Jordan River is a major river in the Middle East, the source of which is in the northern part of Israel. The river flows southward through Israel and then Jordan, emptying into the Dead Sea. East of the river is the collection of districts known as the East Bank.

**GPT cluster #3:** There is no single answer to this question as the east bank of the Jordan River is home to a number of different districts and regions, each with its own unique history, culture, and customs. However, some of the more well-known districts on the east bank include the West Bank, the Gaza Strip, and the Golan Heights.

**Among top-10 generated documents:** Jordan: 10 times Golan Heights: 2 times

Table 19: Case study of retrieved documents and generated documents from language models (4/4).

## B    PROMPTS CHOICES

### B.1    ZERO-SHOT LEARNING PROMPTS (FOR TABLE 1)

#### B.1.1    PROMPTS FOR "INSTRUCTGPT (NO DOCS.)"

We observed the prompts (i.e., "Q: {query}\n\nA:") used in GPT-3 paper (Brown et al., 2020) perform poorly on its text-davinci-002 version. Therefore, we experimented with multiple prompts and found the following two prompts work best on open-domain QA datasets.

– (1) "{query}\n\nThe answer is" (no space between {query} and \n)

– (2) "{query} \n\n The answer is" (performance reported in Table 1)

For fact checking and dialogue system, we used the following prompts.

– Fact Checking "{claim} \n\n Is the claim true or false?"

– Open-domain Dialogue System "{query} \n\n"

#### B.1.2    PROMPTS FOR BACKGROUND GENERATION (STEP-1)

– Open-domain Question Answering "Generate a background document from Wikipedia to answer the given question. \n\n {query} \n\n"

– Fact checking "Generate a background document from Wikipedia to support or refute the statement. \n\n Statement: {claim} \n\n"

– Open-domain Dialogue System "Generate a background document from Wikipedia to answer the given question. \n\n {utterance} \n\n"

#### B.1.3    PROMPTS FOR READING COMPREHENSION (STEP-2)

We collected the prompt from P3 (Bach et al., 2022), which includes over 2,000 open-source prompts for roughly 170 datasets. For zero-shot QA, we experimented with three different reading comprehension prompts. We reported the performance for each prompt in Table 20.

– (1) "Refer to the passage below and answer the following question with just a few words. Passage: {background} \n\n Question: {query} \n\n The answer is"

– (2) "Passage: {background} \n\n Question: {query} \n\n Referring to the passage above, the correct answer (just one entity) to the given question is"

– (3) "Refer to the passage below and answer the following question with just one entity. \n\n Passage: background \n\n Question: query \n\n The answer is"

For fact checking and dialogue system, we chose the simplest prompt from P3.

– Fact Checking "{background} \n\n claim: {claim} \n\n Is the claim true or false?"

– Open-domain Dialogue System "{background} \n\n utterance \n\n"

### B.2    HUMAN PROMPT ANNOTATIONS (FOR SECTION 3.2.1)

In order to get a better prompt for large language models to generate better contextual documents, we asked 30 students in the computer science department to write different prompts. We first constructed a small validation set with 200 examples by combining 50 random question-answer pairs from NQ, 100 random pairs from TriviaQA and 50 random pairs from WebQ. When an annotator wrote down a prompt, our system can immediately evaluate the prompt by using the validation set and return the performance to the annotator. Then, the annotator can modify the previous prompt until the recall performance reaches a threshold, which is set as 50 in our experiments. Finally, we got 29 prompts from human annotators due to two of them are the same. We used the top-10 prompts (shown in Table 21 and Table 22) in the human prompt setting, as described in §3.2.1.

| Models | NQ | | | | TriviaQA | | | | WebQ | | | |
|---|---|---|---|---|---|---|---|---|---|---|---|---|
| | (1) | (2) | (3) | avg. | (1) | (2) | (3) | avg. | (1) | (2) | (3) | avg. |
| *with retriever, AND directly trained on these datasets* | | | | | | | | | | | | |
| DPR + InstructGPT* | 28.8 | 28.5 | 29.9 | 29.1 | 56.0 | 50.1 | 55.3 | 53.8 | 18.9 | 21.7 | 20.1 | 20.2 |
| *with retriever, BUT NOT trained on these datasets* | | | | | | | | | | | | |
| BM25 + InstructGPT | 20.1 | 18.4 | 20.5 | 19.7 | 54.2 | 49.0 | 53.4 | 52.2 | 14.9 | 16.6 | 16.0 | 15.8 |
| Contriever + InstructGPT | 18.3 | 16.5 | 19.1 | 18.0 | 53.1 | 48.5 | 52.4 | 51.3 | 14.9 | 18.2 | 16.8 | 16.6 |
| Google + InstructGPT | **29.1** | **29.3** | 27.8 | **28.8** | **60.3** | 57.5 | 58.7 | 58.8 | 19.5 | 21.8 | 19.9 | 20.4 |
| *without retriever, and not using external documents* | | | | | | | | | | | | |
| GENREAD (InstructGPT) | 27.0 | 28.7 | **28.2** | 28.0 | 58.5 | **59.3** | **59.3** | **59.0** | **22.7** | **26.4** | **24.8** | **24.6** |

Table 20: Zero-shot QA performance under different prompts. The prompts are listed in §B.1.

| No. | Prompts | Validation |
|---|---|---|
| #1 | Generate a background document from Wikipedia to answer the given question. | 66.0 |
| #2 | Provide a background document from Wikipedia to answer the given question. | 65.0 |
| #3 | Generate a background document from web to answer the given question. | 64.0 |
| #4 | Generate a Wikipedia document to support the given question. | 63.5 |
| #5 | Provide a background document for the given question. | 63.0 |
| #6 | Prepare a background document to support the given question. | 63.0 |
| #7 | To support the given question, prepare a background document. | 62.5 |
| #8 | Create a background document that supports the given question. | 61.5 |
| #9 | Retrieve a document from Wikipedia to answer the given question. | 60.5 |
| #10 | Retrieve a Wikipedia article to address the posed question. | 59.5 |

Table 21: Top-10 human prompts, evaluated on merged validation set of NQ, TriviaQA and WebQ.

| Prompt No. | Validation | NQ | WebQ | TriviaQA | Avg. |
|---|---|---|---|---|---|
| #1 (Generate ...) | 66.0 | 45.9 | 51.9 | 68.7 | 55.5 |
| #2 (Provide ...) | 65.0 | 43.9 | 51.0 | 68.3 | 54.4 |
| #3 (Generate ...) | 64.0 | 44.0 | 50.6 | 67.7 | 54.2 |
| #4 (Generate ...) | 63.5 | 43.2 | 51.2 | 67.5 | 54.0 |
| #5 (Provide ...) | 63.0 | 43.6 | 50.3 | 67.9 | 54.0 |
| #6 (Prepare ...) | 63.0 | 43.5 | 50.5 | 67.7 | 54.0 |
| #7 (To support ...) | 62.5 | 43.5 | 50.3 | 67.5 | 53.8 |
| #8 (Create ...) | 61.5 | 42.7 | 50.2 | 66.8 | 53.3 |
| #9 (Retrieve ...) | 60.5 | 41.6 | 49.0 | 68.2 | 53.0 |
| #10 (Retrieve ...) | 59.5 | 40.7 | 49.5 | 67.7 | 52.7 |

Table 22: Performance on NQ, TriviaQA and WebQ test sets of top-10 human prompts.

