# OpenReview forum: "Generate rather than Retrieve: Large Language Models are Strong Context Generators"
_ICLR.cc/2023/Conference — ICLR 2023 poster_

### Official Review · Reviewer_jnAy · 2022-10-23

**Confidence:** 4
**Correctness:** 4
**Technical Novelty And Significance:** 3
**Empirical Novelty And Significance:** 3
**Recommendation:** 8

**Clarity, Quality, Novelty And Reproducibility:**

* The paper is generally well-written and is easy to follow. I have a few questions as I wrote above, but they are relatively minor.
* The quality and the novelty of the paper is above the ICLR standard.
* The authors tried to ensure all experiments are reproducible, including adding results with publicly available LMs. The authors also promised to release the code.


**Strength And Weaknesses:**

### Strengths
* The proposed approach “generate then answer” is new, and is well-executed.
* The paper convincingly shows that the proposed models outperform a range of baselines, and I think this result is quite surprising and new to the research community. The experiments are very solid and extensive, and every competitive baseline I can think of is included in their comparison.

### Weakness
* Given that the findings are very surprising, the key question I have is – why does generate-then-read outperforms retrieve-then-read? And I think this paper does not answer this question, nor tries to answer this question. And I think the paper will significantly be better and more insightful if more analysis that tries to answer this question is provided. For instance, moving some qualitative examples in the appendix to the main text will help. Here are more concrete questions I have (which I don’t think authors should necessarily answer to all of them during rebuttals, but might help answering the question of “why it is the case”):
     * Does it mean that we significantly underestimated LM’s closed-book QA performance, e.g., the LM already has all the knowledge to answer the question but we just didn’t prompt it in the right way, and this type of “elicit” prompting enables the model to get the correct answer more often?
     * Based on examples provided in Appendix (Table 14–17), it looks like the model knows relatively early on what the answer is. For instance in Table 16, all GPT-generated documents already answer the question in their first sentence. Then, why is it significantly better than direct prompting?
     * In fact, based on Table 14–17, I think the generated document has a very different distribution from actual Wikipedia documents. They really look more like a rationale that explains how the answer is arrived, just like chain-of-thought. It would be very interesting to see qualitative/quantitative analysis on how generated documents and retrieved documents are different. For instance, one possible quantitative analysis is to see what happens if the reader trained with retrieved documents is given generated documents, and vice versa.
     * If I aggregate reasonings over the main paper and the appendix, it looks like the paper says retrieve-and-read underperforms because (1) the retrieval accuracy is not good enough due to lacking cross attention, and (2) even if the retrieval was successful, its “readability” is not good since it is written without a question in mind and thus contains information that is not directly related to the question. It would be informative to analyze cases where the proposed model gets correct but retrieve-and-read doesn’t, and see how many of them are (1) or (2).
     * And if a lot of cases turn out to be (2), which the appendix seems to claim – why is finding answers from less readable documents difficult for huge LMs? In other words, how is “recalling knowledge seen during pre-training” easier than “finding the answer from a (less-readable but still answer-containing) document” for a large LM?
* For the clustering-based method, the paper is treating “nucleus sampling” and “diverse human prompts” as a baseline, but isn’t “randomly choosing k question-document pairs” a more obvious baseline and should be included in the main table (Table 2)? Asking because based on Table 10, it looks like a clustering-based method is not really better than random k question-document pairs (~1% difference in R@10 on all three datasets – and  I think the difference will be even less in final answer accuracy). If it is the case, then I am not convinced with the claim in the paper that the clustering-based method is effective, especially given that the clustering-based method is significantly more complicated and expensive (requiring generating documents for all training questions, getting embeddings from GPT-3, etc).
* Writing comment
     * I think the first paragraph of p2 is a bit misleading, because “requiring the model to explain every token in the question and document” is not really shown in the paper (and is even unclear what it means), and “the generated contextual documents tend to be more specific to the question than retrieved documents” is also not really shown in the main paper (although partially shown with a few qualitative examples in the appendix). Also, “Second, large language models have been trained with considerably large amounts of data and parameters, allowing a type of “look up” of world knowledge stored in the model parameters” – seems to be more of a “reason” the method works, rather than the advantage of the method.
     * It would be great to add limitations of the proposed models. For instance, the proposed models may not be able to answer questions about the knowledge LMs are not trained on (since there’s no knowledge to “recall”), may suffer from rare and less popular information, and won’t be able to update its knowledge when there’s a temporal shift (which is the real-world scenario).





#### Side notes
* There is a concurrent paper https://arxiv.org/abs/2210.01296 with a very similar (if not identical) method with different underlying LMs, and shows that generate-then-read does not outperform retrieve-then-read. I am curious what leads to the difference in results, although the methods look very similar. Of course, the authors are not required to discuss this paper in detail given that it is a concurrent (slightly later) work, but I think many potential readers will be curious as well.
* The core idea of the paper is related to the literature in generative retrieval, where a document retrieval is done in a generative manner instead of dense retrieval (Cao et al. “Autoregressive entity retrieval” ICLR, 2021 and a few more). It might be nice to add discussion.
* Question on a zero-shot setting: multiple papers have reported that zero-shot QA accuracy is significantly poor with an Exact Match metric, because the model does not know it should generate a concise text span (e.g., “Olympia”) rather than a sentence (“Zeus was the patron god of the city of Olympia”), and it tends to generate a sentence. This is mentioned in Liu et al. (https://arxiv.org/abs/2103.10385) and a few other papers. I wonder how much of this was the case in the zero-shot setting either in baselines and the proposed models? Was there a special prompt used to prevent this from happening? If “coverage EM” instead of “EM” is used as a metric (counts whether the generated sentence contains the answer or not), is the ranking between models the same?


**Summary Of The Paper:**

This paper proposes a “generate-then-read” approach that contrasts the “retrieve-then-read” approach that has been dominant in open-domain QA. It is based on large language models that encode vast amounts of knowledge and can recall it through prompting. Specifically, the paper proposes two models, one for the zero-shot setting and the other for supervised setting. In the zero-shot setting, the model is simply prompted to generate a relevant document, feeds it back, and generates the answer. In the supervised setting, the model generates a document per training question, clusters them, constructs in-context examples using one question-document pair per cluster, and uses them to generate the document and then the answer as it does in a zero-shot setting. Experiments done on 7 datasets effectively show the model outperforms a range of competitive baselines, including InstructGPT (one of the most competitive large LMs) &  InstructGPT followed by SOTA retrieval systems/Google (in the zero-shot setting) and FiD Followed by DPR/Google (in supervised setting).


**Summary Of The Review:**

* In summary, this is a high quality paper that is definitely above the ICLR standard. The method is new, evaluation is solid and extensive, and results (that the generate-and-read approach outperforms the retrieve-and-read approach) are surprising and teach something new to the research community. The paper is also well written, is easy to follow, and tries to ensure experiments are reproducible.
* As I stated above, the paper does not explain well enough on why generate-and-read achieves such a significant gain (especially compared to retrieve-and-read), which I think is a more important question. It does not include any qualitative examples or analysis in the main text (although it has some in the appendix), which makes it more difficult to understand what is going on with these models.

---

> ### Author Response · Authors · 2022-11-15
> **Response to Reviewer jnAy [5/5]**
>
> > Q6. For the clustering-based method, the paper is treating “nucleus sampling” and “diverse human prompts” as a baseline, but isn’t “randomly choosing k question-document pairs” a more obvious baseline and should be included in the main table (Table 2)? Asking because based on Table 10, it looks like a clustering-based method is not really better than random k question-document pairs (~1% difference in R@10 on all three datasets – and I think the difference will be even less in final answer accuracy). If it is the case, then I am not convinced with the claim in the paper that the clustering-based method is effective, especially given that the clustering-based method is significantly more complicated and expensive (requiring generating documents for all training questions, getting embeddings from GPT-3, etc).
>
> Thank you for raising this question. **In fact, the clustering-based method has almost the same complexity as random k pairs.** Please kindly find our explanations below and we will be very happy to answer your additional questions about this part.
>
> We first note that both clustering-based method or random k pairs are based on the supervised setting, which means we need to generate multiple documents (10 in our experiments) for both training and dev/test sets. For the zero-shot setting, clustering-based method or random k pairs are not appliable since no training set should be given.
>
> **Therefore, both clustering-based method and random k pairs needs to generate 10 times based on 10 different in-context demonstrations, and there is no extra cost on generating documents.** The only difference is how in-context demonstrations are obtained. Besides, generating embeddings can be done under the same decoding pass, and the cost brought by K-means is negligible.
>
> Second, we found that the variance in the performance of random k pairs is very large, due to the process of sampling from the entire dataset. Meanwhile, as generating documents for the entire dataset is computationally intensive, so we re-performed the experiments as follows. We run clustering and random k pairs five times on the dev set, and choose the *median* performance for each setting, then generate 10 documents for both training and dev/test sets, finally train a FiD reader. The performance on end QA is listed below. We updated the Table 10 **(now Table 11 Page 16 in the updated paper)**.
>
> |         |  NQ  | TriviaQA | WebQ |
> |---------|:----:|:--------:|:----:|
> | Random  | 44.1 |   70.9   | 53.3 |
> | Cluster | 45.3 |   71.8   | 54.4 |
>
> Therefore, considering almost the same computational cost, better retrieval and end QA performance, lower performance variance, we believe clustering-based prompts is a better strategy than random k pairs.
>
> Meanwhile, the main goal of our proposed method is to select distinct in-context demonstration as prompts and generate multiple diverse documents that cover different perspectives. Random k pairs also select distinct in-context demonstration as prompts, so the performance of random k pairs (better than sampling but worse than clustering) is consistent with our motivation.
>
>
> > Q7. Concurrent paper https://arxiv.org/abs/2210.01296 with a very similar (if not identical) method with different underlying LMs, and shows that generate-then-read does not outperform retrieve-then-read. I am curious what leads to the difference in results, although the methods look very similar. Of course, the authors are not required to discuss this paper in detail given that it is a concurrent (slightly later) work, but I think many potential readers will be curious as well.
>
> It is great to see other recent works validating ideas similar to ours. We speculate that the difference in performance mainly comes from two aspects. First, the largest model used in their paper is a 62B PaLM, which has much fewer parameters than our 175B GPT-3. Besides, as shown in Fig. 4, we show that scaling with GPT-3 generator parameters can significantly improve the end QA performance, and only the largest GPT-3 can achieve better performance than SoTA retrieve-then-read pipeline, i.e., DPR-FiD. Second, their paper only conducts experiments on few-shot setting, while our paper uses the fully supervised setting.
>
> > Q9. Writing comment and missing references
>
> We really appreciate your suggestions on the details of our paper writing! Based on your suggestions, we now (i) add a limitation section at the end of the paper **(Page 8)**, (ii) revise the first paragraph of page two, and (iii) add the discussions of missing references in the related work section. All of the newly added contents are highlighted in orange.

---

> ### Author Response · Authors · 2022-11-15
> **Response to Reviewer jnAy [4/5]**
>
> **Other of us** think *"less readable documents"* refers to *"all DPR retrieved documents containing the correct answer"*. If so, we think the conclusion of comparison between “recalling knowledge seen during pre-training” and “finding the answer from a (less-readable but still answer-containing) document” cannot be derived from results in **Q4**.
>
> To answer the quesiton *"how is recalling knowledge seen during pre-training easier than finding the answer from a (less-readable but still answer-containing) document for a large LM?*, we first provide the following $2 \times 2$ matrix on three QA datasets.
>
> |    |    |
> |:--:|:--:|
> | TT | TF |
> | FT | FF |
> |    |    |
>
> where
> - TT: Retrieved document has answer (T), and answer is correct (T).
> - TF: Retrieved document has answer (T), but answer is wrong (F).
> - FT: Retrieved document doesn't have answer (F), but answer is correct (T).
> - FF: Retrieved document doesn't have answer (F), and answer is wrong (F).
>
> **Table I:** Performance of baseline: using DPR as retriever and GPT-3 as reader.
>
> |    |     |      |          |      |      |      |     |      |
> |----|:---:|------|----------|------|------|------|-----|------|
> | NQ | 951 |  658 | TriviaQA | 4935 | 1083 | WebQ | 359 |  564 |
> |    | 130 | 1871 |          | 1319 | 3976 |      |  49 | 1060 |
> |    |     |      |          |      |      |      |     |      |
>
> **Table II:** Performance of GenRead: using GPT-3 as generator and GPT-3 as reader.
>
> |    |     |      |          |      |      |      |     |     |
> |----|:---:|------|----------|------|------|------|-----|-----|
> | NQ | 978 |  608 | TriviaQA | 6515 | 1186 | WebQ | 475 | 573 |
> |    |  41 | 1983 |          |  194 | 3418 |      |  28 | 956 |
> |    |     |      |          |      |      |      |     |      |
>
> Now, we think we can use these numbers to answer your questions.
>
> 1. finding the answer from a (less-readable but still answer-containing) document for a large LM
>
>     - It should be TT / (TT + TF) **in Table I**. NQ: 59.1%; TriviaQA: 82.0%; WebQ: 38.9%
>
> 2. finding the answer from a generated document for a large LM
>
>     - It should be TT / (TT + TF) **in Table II**. NQ: 61.7%; TriviaQA: 84.6%; WebQ: 45.4%
>
> 3. recalling knowledge seen during pre-training
>
>     - It should be (TT + TF) / (TT + TF + FT + FF) **in Table II**. NQ: 43.9%; TriviaQA: 68.1%; WebQ: 51.6%
>
> So, we can observe:
>
> - Based on 1 and 2, finding the answer from generated document is *easier* than retrieved document.
>
> - Based on 2 and 3, on the NQ and TriviaQA datasets, finding the answer from a (less-readable but still answer-containing) document is *easier* than recalling knowledge seen during pre-training. However, the observation on the WebQ dataset is the opposite.
>
> In addition, we also provide another observation relevant to your question, which has not been reported in the paper.
>
> 4. finding the answer from a noisy document (not containing the answer)
>
>     - It should be calculated as FT / (FF + FT).
>     - Retrieved documents. NQ: 2.0%, TriviaQA: 5.4%, WebQ: 2.8%
>     - Generated documents. NQ: 6.5%, TriviaQA: 24.9%, WebQ: 4.4%
>
> We can observe that if the retrieved or generated documents do not contain the answer, it becomes more difficult for the LLM to predict the correct answer by reading the generated documents. This is mainly because the generated documents come from the knowledge stored by the model itself, making the model more inclined to determine the answer from the generated documents. Instead, the model might think that there is no answer in the retrieved document and use its own stored knowledge to predict a plausible answer.

---

> ### Author Response · Authors · 2022-11-15
> **Response to Reviewer jnAy [3/5]**
>
> > Q4. If I aggregate reasonings over the main paper and the appendix, it looks like the paper says retrieve-and-read underperforms because (1) the retrieval accuracy is not good enough due to lacking cross attention, and (2) even if the retrieval was successful, its “readability” is not good since it is written without a question in mind and thus contains information that is not directly related to the question. It would be informative to analyze cases where the proposed model gets correct but retrieve-and-read doesn’t, and see how many of them are (1) or (2).
>
>
> **We agree that our proposed method improves (1) both retrieval accuracy (2) readibility. Both automatic metrics (Recall@K in Figure 2 / Table 9, readibility in Table 5) and cases studies (Table 16-19) can reflect it.**
>
> Now, we provide the numbers where the proposed model gets correct but retrieve-and-read doesn’t below, and displat how many of them are (1) or (2).
>
> |     |  NQ | TriviaQA | WebQ |
> |:---:|:---:|:--------:|:----:|
> | (1) | 207 |    462   |  87  |
> | (2) | 270 |    492   |  128 |
>
> > Q5. And if a lot of cases turn out to be (2), which the appendix seems to claim – why is finding answers from less readable documents difficult for huge LMs? In other words, how is “recalling knowledge seen during pre-training” easier than “finding the answer from a (less-readable but still answer-containing) document” for a large LM?
>
>
> **During our discussions, we had some disagreements about understanding the definition of *"less readable documents"* mentioned in your question.**
>
> **Some of us** think that *"less readable documents"* mean *"those documents that are retrieved by DPR but which LLM cannot infer the correct answer from"*. If so, we consider the difficulty of *"finding the answer from a (less-readable but still answer-containing) document"* cannot be quantatively evaluated, since the performance is actually 0. So, the difficulty between *“recalling knowledge seen during pre-training”* and *"finding the answer from a (less-readable but still answer-containing) document"* cannot be compared.
>
>
> At the same time, inferring the correct answer from either retrieved or generated documents is difficult for language models. From Table 5, we observe that when both retrieved and generated documents contain the correct answer, it is *easier* to infer the correct answer from the generated document than the retrieved document. The average improvement on three datasets is 3.7. Despite the significant improvement, there is still an error rate of about 35% on the NQ and WebQ datasets, indicating that inferring the correct answer from the generated documents is still a difficult task.

---

> ### Author Response · Authors · 2022-11-15
> **Response to Reviewer jnAy [2/5]**
>
> > Q2. Based on examples provided in Appendix (Table 14–17), it looks like the model knows relatively early on what the answer is. For instance in Table 16, all GPT-generated documents already answer the question in their first sentence. Then, why is it significantly better than direct prompting?
>
> We consider this "relatively early" observation to be based on comparisons with retrieved documents. This is correct because candidate documents for retrieval is fixed and question-independent, so the answer spans might appear at "relatively late" positions in the document.
>
>
> **However, this is incorrect when compared to direct prompting.** The answer does not always appear in the first sentence of the generated document. We display the distribution of generated documents containing answers in the NQ, TraviaQA and WebQ test sets. We found that only 50% of the answers in NQ and WebQ appeared in the first sentence, reflecting the difficulty of asking LLMs to directly generate the correct answer. Meanwhile, in TriviaQA, the first sentence contains the highest proportion of answers, which is consistent with the observation that LLMs (generally) perform well on TriviaQA dataset.
>
>
> | (tokenization tool: NLTK)     |    NQ          |    TriviaQA                 |   WebQ |
> |-------------------------------|:--------------:|:---------------------------:|:------:|
> | # documents hit the answer    | 16,585         | 77,839                      | 10,784 |
> | # answers in the 1st sentence | 8,532 (51.44%) | 56,035 (71.98%)             | 5,332 (49.44%) |
> | # answers in the 2nd sentence | 2,942 (17.73%) | 10,607 (13.62%)             | 2,166 (20.09%) |
> | # answers in the 3rd sentence | 2,077 (12.52%) | &nbsp; 5,078 &nbsp; (6.53%) | 1,239 (11.49%) |
> | # answers after 3rd sentence  | 3,034 (18.31%) | &nbsp; 6,119 &nbsp; (7.87%) | 2,047 (18.98%) |
>
> Based on this, we think another experiments could also be interesting and provide evidence that generating document-level contexts increases the likelihood of hitting the correct answer.
>
>
> We first extract the subset that generated documents hit correct answers. Then, we report the Cov.EM **(as you suggested in Q8)** of directly generating answers when (i) the answer appears in the first sentence of generated document and (ii) the answer appears after the first sentence of generated document.
>
> | Cov.EM of directly generating answers   |  NQ   | TriviaQA | WebQ  |
> |-----------------------------------------|:-----:|:--------:|:-----:|
> | Answer appears in the first sentence    | 59.2% | 90.5%    | 58.8% |
> | Answer appears after the first sentence | 50.8% | 85.9%    | 49.1% |
>
> From the table we can observe that directly prompting the LLM to generate a short answer has better performance when the answer appears *in the first sentence* than *after the first sentence* of the generated document. This indicates why first generating document-level contexts then inferring an answer from the generating documents is a better solution, especially for examples where correct answers appear *after the first sentence*.
>
> > Q3. In fact, based on Table 14–17, I think the generated document has a very different distribution from actual Wikipedia documents. They really look more like a rationale that explains how the answer is arrived, just like chain-of-thought. It would be very interesting to see qualitative/quantitative analysis on how generated documents and retrieved documents are different. For instance, one possible quantitative analysis is to see what happens if the reader trained with retrieved documents is given generated documents, and vice versa.
>
> We agree that generated document has a different distribution from actual Wikipedia documents. When the reader trained with retrieved documents is given generated documents, or vice versa, the end QA performance drops significantly. Compared with the performance on TriviaQA, the performance drop on NQ and WebQ is more pronounced. We speculate that the reason is that questions in TriviaQA are simpler than NQ and WebQ, reducing the difficulty of inferring the correct answer from retrieved or generated documents.
>
> |                        |      NQ     |   TriviaQA  |     WebQ    |
> |:----------------------:|:-----------:|:-----------:|:-----------:|
> | Retrieved -> Generated | 42.2 (-3.4) | 70.1 (-1.4) | 51.4 (-3.0) |
> | Generated -> Retrieved | 46.6 (-3.5) | 64.5 (-1.7) | 47.9 (-2.9) |

---

> ### Author Response · Authors · 2022-11-15
> **Response to Reviewer jnAy [1/5]**
>
> Thanks a lot for your reviews! Your professional reviews offer us great advice towards writing a more comprehensive and competitive paper! And, we are very encouraged that you found our method is novel and well-executed!
>
>
> > Q8. Question on a zero-shot setting: multiple papers have reported that zero-shot QA accuracy is significantly poor with an Exact Match metric, because the model does not know it should generate a concise text span (e.g., “Olympia”) rather than a sentence (“Zeus was the patron god of the city of Olympia”), and it tends to generate a sentence. This is mentioned in Liu et al. (https://arxiv.org/abs/2103.10385) and a few other papers. I wonder how much of this was the case in the zero-shot setting either in baselines and the proposed models? Was there a special prompt used to prevent this from happening? If “coverage EM” instead of “EM” is used as a metric (counts whether the generated sentence contains the answer or not), is the ranking between models the same?
>
> **We answer the question about Coverage EM (Cov.EM) first because we will refer to it in Q2, so first to clarify its definition.**
>
> We agree with the observations of existing papers such as Liu et al, and your suggestion on adding "Coverage EM" as an evaluation metric. We define "Coverage EM" (short as Cov.EM) as the percentage of generated outputs that **contain** the answer.
>
> We first display the baseline performance of directly asking GPT-3 to generate an answer. The EM scores are copied from Table 2.
>
> | Baseline |   NQ   | TriviaQA |  WebQ  |
> |----------|:------:|:--------:|:------:|
> | EM       |  20.90 |   52.60  |  18.60 |
> | Cov.EM   |  29.61 |   64.75  |  33.81 |
>
> The issue could also occur when using our generate-then-read pipeline, becuase the output from the reading step might also contain the answer but not the exact answer span.
>
> We now display the performan of our GenRead. The EM scores are copied from Table 2.
>
> | GenRead  |   NQ   | TriviaQA |  WebQ  |
> |----------|:------:|:--------:|:------:|
> | EM       |  28.23 |   59.30  |  24.75 |
> | Cov.EM   |  37.78 |   66.43  |  38.88 |
>
> Therefore, the ranking between models is the same! We tried different prompts for directly generating answers, including (i) our own prompts (ii) prompts from the GPT-3 paper and (iii) prompts from P3 [1], in our baseline implementation, but actually found the one reported in GPT-3 paper worked best.
>
> [1] PromptSource: An Integrated Development Environment and Repository for Natural Language Prompts. ACL 2022. https://arxiv.org/abs/2202.01279
>
>
> > Q1. Does it mean that we significantly underestimated LM’s closed-book QA performance, e.g., the LM already has all the knowledge to answer the question but we just didn’t prompt it in the right way, and this type of “elicit” prompting enables the model to get the correct answer more often?
>
> **We believe the answer is yes**! Existing methods directly prompt large language models (LLMs) to generate short answers (under a single decoding pass), limiting the performance of LLMs on the close-book QA task. We illustrate why our pipeline, i.e., first generating **multiple** (reason #2) **document-level contexts** (reason #1) then inferring an answer form the generating documents, is a better solution.
>
>
> - Reason #1: During pre-training of LLMs, document-level text is the dominant type of input and output. So, the task of directly generating short answers **prevents** LLMs from effectively recalling knowledge from their memory. However, the goal of generating document-level context for a given question can serve as an intermediate step that simulate the task of language modeling pre-training, which is closer to the objective of language model pre-training than directly generating short answers.
>
> - Reason #2: Multiple decoding passes can increase the likelihood of generated documents hitting the correct answer (i.e., better recall). Directly generating the answer can also produce multiple outputs, but it can be difficult to determine which is the correct one. Instead, the generated documents provide question-specific justifications, and using a trained reader model could efficiently identify the answer across multiple generated documents.

---

### Official Review · Reviewer_a9DF · 2022-10-24

**Confidence:** 4
**Correctness:** 4
**Technical Novelty And Significance:** 4
**Empirical Novelty And Significance:** 4
**Recommendation:** 10

**Clarity, Quality, Novelty And Reproducibility:**


The only downside of this study is that although all the used models are publicly available, API costs or amount of resources needed,  replicating the experiments or following up on this idea, is going to be a privilege of an handful of universities/companies.



**Details Of Ethics Concerns:**

Mild concerns about racial/gender/... bias in large language models.

**Strength And Weaknesses:**

The experimental setting is clear and sound, there are many ablation study and experiments that give insights on the tradeoff input docs and model size.
Related work is well covered.

I appreciate the study on the performance/model size tradeoff. Figuring out if emergent abilities can be pushed on smaller models (like has been done with PET for few shot learning), maybe using different prompting or light finetuning etc. it is an interesting research direction.

In the introduction the authors claim ther are 3 drawbacks of retrieval models, but apart for point 3 I don't see the proposed approach to directly address the first two points. Please, clarify this.

page 5 bottom of the page Gppher -> Gopher


**Summary Of The Paper:**

The paper propose a new paradigm to solve the task of question answering.
In contrast of the standard paradigm retrieve and read, where first a set of documents are retrieved according to a question, and then they are "read" to answer the question, the proposed model first generates documents relevant to a question with a large language model, and then a reader answer the question based on the generated documents.

The authors propose a clustering approach to diversify the generated documents to increase recall.

The model is tested in two settings, where the reader is a large language model, and is a fine tuned QA system.
Results on three tasks and six benchmarks show that the approach is very promising, outperforming standard retrieve and read sota models.


**Summary Of The Review:**

The paper is sound and I don't see any major points for it to get rejected.
It is a novel idea, well executed and well presented.

---

> ### Author Response · Authors · 2022-11-15
> **Response to Reviewer a9DF [1/1]**
>
> Thank you very much for reviewing and supporting our paper! We are very encouraged that you found our paper is novel, well executed and well presented!
>
> > Q1. In the introduction the authors claim there are 3 drawbacks of retrieval models, but apart for point 3 I don't see the proposed approach to directly address the first two points. Please, clarify this.
>
> We are sorry for making this confusion! We slightly modified the writings of the first two drawbacks to make them clearer. They are highlighted in orange in the revised paper.
>
> We talked about the drawbacks of the retrieval method in the first paragraph, with the aim of presenting the motivation for our method **"generate rather than retrieve"**.
>
> First, candidate documents for retrieval are fixed, so the retrieved documents might contain noisy information irrelevant to the question. Second, the representations of questions and documents are typically obtained independently in modern two-tower dense retrieval models, leading to only shallow interactions captured between them.
>
> These two drawbacks limit the performance of retrieving most relevant documents for the given question. In contrast, large language models can naturally overcome both drawbacks, as they generate contextual documents conditioned on the input question, and perform deep token-level cross-attention between the question and document (already generated), resulting in the generated documents to be more specific to the question than the retrieved documents.
>
> > Q2. The only downside of this study is that although all the used models are publicly available, API costs or amount of resources needed, replicating the experiments or following up on this idea, is going to be a privilege of an handful of universities/companies.
>
>
> We agree with your concerns in this regard. While current research on large language models is somewhat limited by computational resources, we believe that this will be gradually solved in the future.
>
> - The price of OpenAI GPT-3 (text-davinci-002) has been reduced to 0.02 USD per 1,000 tokens, making it very cheap to reproduce the experiments from our paper and conduct other experiments. For example, in our experiments, the average token per generated document is 55. So for every 1 USD spent, you can generate one context document each for 1,000 questions.
>
>
> - Some optimization libraries, such as DeepSpeed, can now significantly reduce memory usage, and run large language models with better parallelism on existing computer hardware. In our experiments, we used 8 x A100 GPUs to run OPT-175B, which could probably already be runnable in some university labs, though still a bit expensive.
>
>
> - The NLP community is increasingly interested in training better large language models. With better training strategies, some medium-sized models (with parameters between 10B and 100B) are likely to perform well on such tasks in the future.
>
>
> > Writing: page 5 bottom of the page Gppher -> Gopher
>
> We revised the typo. Thank you for pointing it out!

---

### Official Review · Reviewer_2jcY · 2022-10-25

**Confidence:** 4
**Correctness:** 3
**Technical Novelty And Significance:** 3
**Empirical Novelty And Significance:** 3
**Recommendation:** 8

**Clarity, Quality, Novelty And Reproducibility:**

The paper is easy to follow and the motivation is clear. There are some concerns about reproducibility, while this is not a major issue if the code is available.

**Strength And Weaknesses:**

Strength:
1. The proposed method to generate whole contextual documents effectively utilize the internal knowledge stored in a large language model, without a retrieval process from external sources.
2. Experiments on three knowledge-intesive tasks show the strong performance of GENREAD on both zero-shot and supervised settings.
3. The authors further provide the reproducibility results with open-source large langue models which is beneficial for future research.

Weakness:
1. Although experiments show the effectiveness of generation, more analysis on the need of generation would be helpful.
- In Table 2 and 3, merging the retrieved documents with generated documents further enhance the performance. Which kinds of knowledge can be additionally captured by internal knowledge (generation) that might be missed during retrieval?
2. Generation with large language models can suffer from hallucination, while there is no discussion on this aspect. Qualitative analysis of the generated documents would be interesting.
3. Generating whole documents might require more computational cost than retrieval during inference. However, there is no discussion on inference time or the cost in the paper.
4. The zero-shot performance can be changed according to the choice of prompts, while the details on the choice and the performance variance with different prompts are not provided.


**Summary Of The Paper:**

This paper proposes the generate-then-read (GENREAD) approach to solve knowledge-intensive tasks. Compared to the previous retrieve-then-read approach, the proposed method first generates the contextual documents by prompting a large language model. The authors provide two variants of reader, zero-shot and supervised setting, and prove the effectiveness of generation in both settings.

**Summary Of The Review:**

In my opinion, this paper proposes a novel approach for knowledge-intesive tasks, replacing retrieval with generation. GNEREAD proves its effectiveness in both zero-shot and supervised settings. The experiment results are interesting while more discussions about the need of generation and generated documents would give more intuition.

---

> ### Author Response · Authors · 2022-11-14
> **Response to Reviewer 2jcY [2/2]**
>
> > Q3. Generating whole documents might require more computational cost than retrieval during inference. However, there is no discussion on inference time or the cost in the paper.
>
>
> **Thank you for your suggestion. We added the following discussion in Section A.6 (Page 17) in Appendix.**
>
> We use the FLOPs-per-token estimates for Transformer-based language models, which is introduced by [4].
> We consider DPR using the BERT-base version with 110M parameters and GPT-3 using its largest version with 175B parameters. For the DPR model, all Wikipedia documents only need to be encoded once. Therefore, as the number of input questions increases, the marginal computational cost gradually decreases. For fair comparison, we first use DPR to encode all 21M Wikipedia documents once. Encoding all Wikipedia documents requires $110e6$ (BERT-base parameters) $\times$ $21e6$ (total number of documents) $\times$ $100$ (tokens per document) $= 2.3e17$ FLOPs. When the embedding of all candidate documents are produced, retrieving documents for a given question requires $110e6$ (BERT-base parameters) $\times$ $20$ (tokens per question) $+$ $21e6$ (total number of documents) $\times$ $(768 + 768 - 1)$ $= 3.2e10$ FLOPs.
>
> So, the equation for the total cost $Y_{\text{DPR-cost}}$ to retrieve documents using DPR versus the number of input questions $X$ is:
>
> - $Y_{\text{DPR-cost}} = 3.2e10 X + 2.3e17 $
>
> For the GPT-3 model, it requires $\times$ $175e9$ (GPT-3 parameters) $\times$ $10$ (number of documents) $\times$ $55$ (generated tokens per document) $=$ $5.5e13$ FLOPs to generate contextual documents for a given question.
>
> So, the equation for the total cost $Y_{\text{GPT3-cost}}$ to generate 10 documents using GPT-3 versus the number of input questions $X$ is:
>
> - $Y_{\text{GPT3-cost}} = 9.6e13 X $
>
> When $Y_{\text{DPR-cost}} = Y_{\text{GPT3-cost}}$, $X \approx 2473 $
>
> In conclusion, if the number of input questions is less than $2473$, the total cost of GPT-3 is lower than the DPR; if the number of input questions is greater than $2473$, the total cost of GPT-3 exceeds the DPR.
>
>
> [4] Scaling Laws for Neural Language Models. 2020 https://arxiv.org/abs/2001.08361
>
> > Q4. The zero-shot performance can be changed according to the choice of prompts, while the details on the choice and the performance variance with different prompts are not provided.
>
> Thank you for pointing it out! We collected the prompt from P3 [5], which includes over 2,000 open-source prompts for roughly 170 datasets.
>
> For zero-shot QA, we experimented with three different reading comprehension prompts. We show the average performance (EM score) for three prompts below, with standard deviation in parentheses.
>
> |                 |      NQ     |   TriviaQA  |     WebQ    |
> |-----------------|:-----------:|:-----------:|:-----------:|
> | DPR+GPT3        | 29.1 (0.60) | 53.8 (2.63) | 20.2 (1.15) |
> | BM25+GPT3       | 19.7 (0.91) | 52.2 (2.29) | 15.8 (0.70) |
> | Contriever+GPT3 | 18.0 (1.09) | 51.3 (2.02) | 16.6 (1.35) |
> | Google+GPT3     | 27.8 (0.21) | 58.8 (1.15) | 20.4 (1.00) |
> | GenRead(GPT-3)  | 28.0 (0.71) | 59.0 (0.38) | 24.6 (1.52) |
>
> **We added the performance for each prompt in Table 19 (Page 25). We also listed the choices of prompts of QA and other KILT tasks in Appendix B.1 (Page 25).**
>
> Besides, we wanted to claim that all models in Table 1 are fairly compared because they used the same prompts. The only difference is the source of documents, e.g., (i) no document, (ii) documents from BM25/DPR/Google, (iii) documents from GPT.
>
> [5] Promptsource: An integrated development environment and repository for natural language prompts. ACL 2020 https://arxiv.org/abs/2202.01279

---

> ### Author Response · Authors · 2022-11-14
> **Response to Reviewer 2jcY [1/2]**
>
> Thanks a lot for your reviews! Your professional reviews offer us great advice towards writing a more comprehensive and competitive paper! And, we are very encouraged that you found our method is novel and interesting!
>
> > Q1. Although experiments show the effectiveness of generation, more analysis on the need of generation would be helpful. In Table 2 and 3, merging the retrieved documents with generated documents further enhance the performance. Which kinds of knowledge can be additionally captured by internal knowledge (generation) that might be missed during retrieval?
>
> Thanks for your advice! We first illustrate the need for generation, followed by a quantitative analysis.
>
> First of all, the reason why the retrieve-then-read pipeline produces wrong answers is mainly divided into two aspects:
>
> - Retrieval error: the retrieved documents do not contain the correct answer;
> - Reading errors: the reader cannot infer the correct answer from retrieved documents.
>
> **We consider that internal knowledge (generation) could bring improvements in both of the above-mentioned aspects.**
>
> - The generated documents contain the correct answer more frequently than the top retrieved documents using a dense retriever, which is reflected in better Recall@K scores (Figure 2). Because dense retrieval relies on question-document similarity to rank candidate documents, answers may be far away from parts of the document that are similar to the question, without being included in the current chunk.
>
> - The generated documents contain more question-specific justifications, making it easier for readers to infer answers from them. In contrast, dense retrievers encode question and context documents separately, resulting in retrieved documents that are less specific to the question than generated documents conditioned on the question text as input to the generator.
>
> Second, we show the percentage of retrieval errors and read errors for DPR-FiD on NQ, TriviaQA, and WebQ datasets.
>
> |                  |      NQ     |   TriviaQA   |     WebQ    |
> |------------------|:-----------:|:------------:|:-----------:|
> | Retrieval errors | 873 (48.7%) | 2548 (66.7%) | 490 (49.6%) |
> | Reading errors   | 921 (51.3%) | 1272 (33.3%) | 497 (50.4%) |
>
> Then, we show the percentage of errors that can be resolved by merging the retrieved documents with the generated documents.
>
> |                                 |        NQ       |     TriviaQA     |       WebQ      |
> |---------------------------------|:---------------:|:----------------:|:---------------:|
> | % of Retrieval errors resolved  | 12.7% (111/873) | 29.6% (753/2548) | 16.3% (80/490)  |
> | % of Reading errors resolved    | 23.8% (219/921) | 39.4% (501/1272) | 26.6% (132/497) |
>
>
> From the tables, we can observe that both retrieval error and reading error cases can be partly resolved, after merging retrieved and generated documents. Besides, more reading errors are resolved than retrieval errors, indicating the generated documents make it easier for the reader to infer the correct answer from them. **This can also be observed from our case studies in Table 16-19 (Page 20) in the appendix.**
>
>
> > Q2. Generation with large language models can suffer from hallucination, while there is no discussion on this aspect. Qualitative analysis of the generated documents would be interesting.
>
> We agree that hallucination is commonly shared by many natural language generation tasks [1], which could also occur in our contextual document generation process. We admit that our proposed clustering-based prompt is not designed for alleviating hallucinations during generation. Consideration in combination with recent approaches [2] to boost generative faithfulness is a direction worthy of future research. **We added discussions in our future work in Page 8.**
>
> As there is no standard automatic metric to evaluate hallucination [1]. So, we added a human study to evaluate the generated documents based on "factuality", where factuality refers to the quality of being actual or based on world knowledge [1,3].
> Our evaluation is based on 100 examples in the NQ dataset. As shown in Table 9, there are 8 documents with hallucinations, and **we demonstrate the cases in Table 15 (Page 20).**
>
> For example, in Table 15 (Page 20), the generated document for ''Who died in the first episode of Stranger Things?'' is ''In the first episode of Stranger Things, the character Will Byers dies. He is killed by the Demogorgon, a monster from the Upside Down.'' This is actually incorrect because Will's death was faked in Season 1, and he is still alive at the end of season 1, but GPT-3 thought he was dead.
>
> [1] Survey of Hallucination in Natural Language Generation. 2022. https://arxiv.org/abs/2202.03629
>
> [2] Faithful Reasoning Using Large Language Models. 2022. https://arxiv.org/abs/2208.14271
>
> [3] On Faithfulness and Factuality in Abstractive Summarization. ACL 2020. https://arxiv.org/abs/2005.00661

---

### Official Review · Reviewer_Vpy9 · 2022-10-26

**Confidence:** 4
**Correctness:** 3
**Technical Novelty And Significance:** 2
**Empirical Novelty And Significance:** 3
**Recommendation:** 8

**Clarity, Quality, Novelty And Reproducibility:**

Clarity: The paper is clearly written overall. Its a simple idea and the paper communicates that well.

Quality: There is a growing body of recent work that are increasingly using LLMs to generate knowledge instead of retrieving from external knowledge sources. This paper is a work in that line and I would say overall the positive results puts the paper as high quality. I am personally not sure about the diverse retrieval claim as it can be verified more with evaluation on bechmarks such as Qampari and on multi-hop questions. The current results in Table 4 regarding diversity does not fully convince me that the procedure leads to diverse results.

Originality: The technical novelty of the work is limited. However, I view the simplicity of the model combined with good empirical results as a positive contribution.

Reproducibility: GPT-3 API is behind a paywall, so reproducibility is not straightforward. I appreciate the author's experiments on the open-source OPT models, however, the performance seems to be lacking when OPT is used.

**Strength And Weaknesses:**

Strengths:

- There is a growing body of recent work that is increasingly using LLMs to generate knowledge instead of retrieving it from external knowledge sources. This paper proposes a model in that line of work that is fairly interesting and obtains good experimental results.
- The proposed model gets promising results on open-domain QA, fact-retrieval, and on wizard-of-Wikipedia (open-domain dialogue) tasks. The performance of just the model is comparably or modestly better than retrieve-and-read approaches, but combining evidence from retrieval and generation leads to nice gains on tasks, suggesting that the proposed approach is practical.
- The paper is overall well-written and easy to understand

Weakness:

- A weakness of the proposed model is that the clustering and selecting of the prompts is *independent* of the given question. Essentially, if the same n (question, document) pairs are selected from each cluster, the prompts are exactly the same for each question. This might be suboptimal. For example, do you have a sense of how many prompts on average are useful for retrieval for a given question? It would also be interesting to devise question-dependent prompting.
- Although I like the idea of the power of LLMs to generate highly contextual evidence for a given question, I believe the paper would be stronger if it included two more evaluation
    - Since the paper focuses on the LLM's ability to do diverse retrieval, it would be good to test that on a benchmark that is explicitly designed to test that ability. For example, Qampari (Amouyal et al 2022) is a recent benchmark that tests for questions with multiple answers. It would be interesting to see the results on that.
    - Multi-hop questions: I can imagine that LLMs might have the ability to generate contextual evidence that fuses information from two or more documents required to answer a multi-hop question. I think that might be a powerful way for answering multi-hop questions and it would be nice if the authors report some results on whether that is happening or not.
- In Fig 2, the difference between clustering and sampling method doesn't seem to be too much. Could you report (in numbers) what is the exact difference in recall %? Also, just to clarify, does “sampling” refers to using nucleus sampling to generate K different documents (and not using other query-doc pairs in the prompt)? Is that correct? If so, I am a little worried about the very less difference between a method that does not use different (q,d) pairs in prompts and a method that uses.
- I am confused by Table 4. What is the metric used since the values seem to be between 0.84 and 2.17? My guess is it is the average number of answers found by the model. If so, couple of comments:
    - The difference between DPR and the proposed method is not that high (1.9 v/s 2.17 in TriviaQA and 1.1 v/s 1.22 in WQ; in NQ, DPR actually outperforms)
    - I believe the table would be more readable if % coverage is reported and that would also help us to understand how much is the difference in performance between DPR. This is important and I would like to see this in the author rebuttal.
- In Table 5, do you have any more insights as to why is the performance of OPT so behind that of GPT3 (e.g. in TriviaQA)
- I would like to have the authors add an additional section regarding the failure modes of the model. For example:
    - What happens when the evidence generated by the LLMs consists of incorrect factual information? For example, in Table 17, the LLM generates "The collection of the districts to the east of the Jordan River is known as the West Bank." - this I believe is factually incorrect. Since Westbank is actually west of the Jordan River. To the east of the Jordan river is Jordan, which is the gold answer. Did the reader model still read this and output the right answer?
    - Can adding evidence also from the retriever (i.e. real documents) help in mitigating some of the possible drawbacks that might arise because of LLMs hallucinating incorrect information?
- In sec 3.1 where the joint probability of p(a,d | q) is described, I believe a summation of d is missing (marginalization).
- “due to the high coincidence of token distributions in the decoding process under a single prompt.” - What does this exactly mean?
- [Minor] - I think the related work section can be written better. Currently, it reads like a list of related work and there are no explicit points differentiating wrt current work. Currently, it only mentions that this work is the first work to apply it for knowledge-intensive tasks which, I believe is actually underselling this work.

**Summary Of The Paper:**

This paper presents a generate-and-read approach for solving knowledge-intensive tasks. Different from the retrieve-and-read approach that first retrieves evidence (paragraphs) from a corpus before feeding it to reader models, the current approach uses large language models (LLMs) to generate contextually relevant evidence which is then fed to a reader model.

To generate multiple diverse contextual evidence, the paper proposes a clustering-based prompting technique where the prompt is synthesized from (question, evidence) pairs from diverse clusters. Specifically, for each query in the training set, the top-1 document is retrieved using a standard retriever such as BM-25 or by prompting an LLM to generate a document. Next, the query-doc pairs are encoded with an LLM (GPT-3) to obtain 12,288-dimensional vectors. These vectors are then clustered using k-means. Next, the “n” query-document pair from each of these clusters form an in-context example which is used to generate a paragraph. This step is repeated k-times for each cluster to generate “k”-diverse documents. In contrast, retrieve-and-read approaches often do not enjoy this type of control during the retrieval process and likely retrieve multiple redundant paragraphs sacrificing diversity.

The proposed approach was evaluated on several knowledge-intensive tasks such as open-domain QA, fact-retrieval, and on wizard-of-Wikipedia (open-domain dialogue) tasks.

The proposed method achieves marginal improvement over retrieve-and-read baselines and significant improvements when the evidence from both retrieval and generation are combined.

**Summary Of The Review:**

The paper proposes a simple and effective way of generating relevant contextual evidence for knowledge-intensive tasks. The proposed method achieves marginal improvement over retrieve-and-read baselines and significant improvements when the evidence from both retrieval and generation are combined. I am not fully convinced about the diverse retrieval (one of the primary motivations for the paper) and would like to see more experiments and clarification. Hence I am leaning toward weak acceptance.

---

> ### Author Response · Authors · 2022-11-15
> **Response to Reviewer Vpy9 [4/4]**
>
> > Q6. In Table 5, do you have any more insights as to why is the performance of OPT so behind that of GPT3 (e.g. in TriviaQA)
>
> We are not absolutely sure about the reason, but we speculate that this lag is due to the difference in hyperparameters between OPT and GPT-3 during pre-training. As reported in Figure 4 in the OPT paper [7], OPT lag behinds GPT-3 on average performance across 14 NLP tasks. Therefore, the difference in our QA performance is likely due to this as well.
>
> [7] OPT: Open Pre-trained Transformer Language Models. 2022. https://arxiv.org/abs/2205.01068
>
> > Q7. I would like to have the authors add an additional section regarding the failure modes of the model. For example: What happens when the evidence generated by the LLMs consists of incorrect factual information? For example, in Table 17, the LLM generates "The collection of the districts to the east of the Jordan River is known as the West Bank." - this I believe is factually incorrect. Since Westbank is actually west of the Jordan River. To the east of the Jordan river is Jordan, which is the gold answer. Did the reader model still read this and output the right answer? Can adding evidence also from the retriever (i.e. real documents) help in mitigating some of the possible drawbacks that might arise because of LLMs hallucinating incorrect information?
>
> Yes. The generated document in the example you mentioned has hallucinations. We agree that hallucination is commonly shared by many natural language generation tasks [8], which could also occur in our contextual document generation process.
>
> As there is no standard automatic metric to evaluate hallucination [8], we cannot count the total number of *“adding evidence from the retriever help in mitigating the drawbacks that LLMs hallucinate incorrect information”* among all predictions. So, we added a human evaluation to study the failure cases of the model. Our evaluation is based on 100 examples in the NQ datasets. As shown in Table 9, there are 8 wrong predictions caused by hallucination, and **we demonstrate the cases in Table 15 (Page 20).** We found only one answer is corrected by merging documents from both sources. This is might because for these questions contain entities with relatively low frequency, making retrieval model also hard to find golden evidence from Wikipedia.
>
> Finally, consideration in combination with recent approaches [9] to boost generative faithfulness is a direction worthy of future research. **We added discussions in our future work on Page 9.**
>
> [8] Survey of Hallucination in Natural Language Generation. 2022. https://arxiv.org/abs/2202.03629
>
> [9] Faithful Reasoning Using Large Language Models. 2022. https://arxiv.org/abs/2208.14271
>
> > Q8. “due to the high coincidence of token distributions in the decoding process under a single prompt.” - What does this exactly mean?
>
> We realized this expression might cause some confusions. We revised it as "Compared to dense retrievers, simply prompting a large language model to generate multiple contextual documents often leads to low knowledge coverage, **since the contents generated by multiple decoding passes from the same prompt and input question tend to be similar.**"
>
> > Q9. [Minor] - related work.
>
> Thank you! We revised the related work and highlighted the revision in the updated version.

---

> ### Author Response · Authors · 2022-11-15
> **Response to Reviewer Vpy9 [3/4]**
>
> > Q4. Multi-hop questions: I can imagine that LLMs might have the ability to generate contextual evidence that fuses information from two or more documents required to answer a multi-hop question. I think that might be a powerful way for answering multi-hop questions and it would be nice if the authors report some results on whether that is happening or not.
>
> Thanks for your suggestion! We added two experiments on HotpotQA, including (i) comparing with **GPT-3** that directly generates an answer, and (ii) comparing with **single-hop / multi-hop DPR models** that retrieve documents from Wikipedia.
>
>
> **Table I:** QA performance of GPT-3 and GenRead, evaluated by exact match (EM) score.
>
> | EM score | HotpotQA |
> |----------|:--------:|
> | GPT-3    |   20.7   |
> | GenRead  |   25.6   |
>
>
> **Table II:** Recall@10 for generated documents by GPT-3 and retrieved documents by DPR.
>
> |    Recall@10      | HotpotQA |
> |-------------------|:--------:|
> | Single-hop DPR [3]|   45.4   |
> | Multi-hop DPR [4] |   77.5   |
> | Ours (sampling)   |   52.5   |
> | Ours (clustering) |   55.1   |
>
> [3] Dense Passage Retrieval for Open-Domain Question Answering. EMNLP 2020. https://arxiv.org/abs/2004.04906
>
> [4] Answering Complex Open-Domain Questions with Multi-Hop Dense Retrieval. ICLR 2021. https://arxiv.org/abs/2009.12756
>
> From the Table I, we can observe GenRead can significantly outperform GPT-3 by a large margin, which is consistent with the performance on single-hop QA setting **(Table 1 on page 5 of the main paper)**.
>
> From the Table II, we can observe
>
> - Compared to single-hop retrieval (55.1 > 45.4), the generated documents contain the correct answer more frequently than the top retrieved documents using a single-hop retriever. This indicates LLMs have the ability to generate contextual evidence that fuses information from multiple documents required to answer a multi-hop question.
>
> - Compared to multi-hop retrieval (55.1 < 77.5), the Recall@10 of generated documents is significantly behind, indicating that our simple prompting method is not good enough on complex QA scenarios.
>
>
> **We note that two concurrent papers work on improving LLM for complex QA scenarios via decomposed prompting, chain-of-thought reasoning, and self-consistency technique with multiple-path decoding [5,6].**
>
> Both papers report the performance of HotpotQA, showing that multi-hop QA requires a more complex reasoning process. However, neither papers can achieve better performance than *retrieve-then-read* pipeline, as we did on the single-hop QA and fact checking tasks.
>
> Instead, the main contribution of our paper is we proposed a simple generate-then-read pipeline that uses a novel clustering-based prompting approach to generate multiple diverse contextual documents. Our method can outperform *retrieve-then-read* pipeline, achieving SoTA performance on single-hop QA benchmarks, without retrieving any documents from any external knowledge source.
>
> [5] Recitation-Augmented Language Models. Under review at ICLR 2023. https://openreview.net/forum?id=-cqvvvb-NkI
>
> [6] Decomposed Prompting: A Modular Approach for Solving Complex Tasks.  Under review at ICLR 2023. https://openreview.net/forum?id=_nGgzQjzaRy
>
>
>
> > Q5. In Fig 2, the difference between clustering and sampling method doesn't seem to be too much. Could you report (in numbers) what is the exact difference in recall %? Also, just to clarify, does “sampling” refers to using nucleus sampling to generate K different documents (and not using other query-doc pairs in the prompt)? Is that correct? If so, I am a little worried about the very less difference between a method that does not use different (q,d) pairs in prompts and a method that uses.
>
> Yes. It means using nucleus sampling to generate K different documents under the same prompt (and not using other query-doc pairs in the prompt). The prompt selection is discussed in Tables 12-13.
>
> Beside, for all numbers in Figure 2, we also put them in Table 10 in Appendix (we added this in the Figure 2 caption, to help readers refer to the exact numbers).
>
> **The numbers are copied from Table 10 on Page 16.**
>
> | Recall@10         |  NQ  | TriviaQA | WebQ |
> |-------------------|:----:|:--------:|:----:|
> | Ours (nucleus)    | 66.2 |   81.6   | 71.4 |
> | Ours (clustering) | 70.9 |   82.9   | 73.3 |
>
> Because the Y-axis span of Figure 2 is too large (from 50 to 90), it seems that the improvement is not obvious, but according to the numbers in Table 10, Recall@10 has an average improvement of +2.63 on the three open-domain QA datasets. Meanwhile, according to Table 2, the exact match of end-QA is improved by +2.40 on average (FiD-770M as reader). Therefore, we think that clustering-based prompt makes significant improvements over nucleus sampling under the same prompt.

---

> > ### Comment · Reviewer_Vpy9 · 2022-12-05
> > **Thank you for adding the results for HotpotQA**
> >
> > I thank the authors for adding the results for multi-hop QA benchmark. The results clearly show that the proposed method is better than single-hop methods but lag behind multi-hop methods - which is reasonable. I think the paper would be stronger if these results are added to the paper.

---

> > > ### Author Response · Authors · 2022-12-07
> > > **Response to results on HotpotQA**
> > >
> > > Thanks for your suggestion! We will add the results to the paper, also discuss the inadequacy of our method on multi-hop QA and point out the future direction of addressing multi-hop QA through complex reasoning.

---

> ### Author Response · Authors · 2022-11-15
> **Response to Reviewer Vpy9 [2/4]**
>
> > Q2. I am confused by Table 4. What is the metric used since the values seem to be between 0.84 and 2.17? My guess is it is the average number of answers found by the model. If so, couple of comments: (1) the difference between DPR and the proposed method is not that high (1.9 v/s 2.17 in TriviaQA and 1.1 v/s 1.22 in WQ; in NQ, DPR actually outperforms) (2) I believe the table would be more readable if % coverage is reported and that would also help us to understand how much is the difference in performance between DPR. This is important and I would like to see this in the author rebuttal.
>
> > Q3. Since the paper focuses on the LLM's ability to do diverse retrieval, it would be good to test that on a benchmark that is explicitly designed to test that ability. For example, Qampari (Amouyal et al 2022) is a recent benchmark that tests for questions with multiple answers. It would be interesting to see the results on that.
>
> We are sorry for making the confusions, and thank you for suggesting the Qampari dataset [2]. We answer two questions together, and add experiments on both existing datasets already used in our paper (NQ, TriviaQA, WebQ), and the Qampari dataset.
>
> In the Table 4, the metrics is how many unique answers contained in the generated documents. For example, WebQ questions have 2.39 answers in average, and DPR documents contain 1.10 correct answers in average. The mininum value of the metric is 0, the the maxinum value is the average answer for each question (i.e., 2.39 for WebQ).
>
> We agree with your suggestion, and consider the coverage as a better metric for evaluation. So we define the Coverage@K as
>
> - Coverage@K = $\frac{1}{n} \sum_{i}^{n} \frac{\text{Number of hit answers for i-th question in top-K documents}}{\text{Number of all possible answers for i-th question}}$
>
> where $n$ is the total number of questions in a dataset. **We note that when the question has only one answer, Coverage@K is the same as Recall@K.**
>
> So now, we display the Coverage@10 on NQ, TriviaQA, WebQ and QAMPARI datasets. We also note that the TriviaQA dataset considers entity alias (e.g., "United States" has "USA", "U.S" and etc) as separate answers. We report two versions in the table, (i) not including alias and (ii) including alias. The numbers of TriviaQA (with alias) and QAMPARI are low because TriviaQA has 14.02 answers in average (when including alias) and QAMPARI has 12.5 answers in average.
>
> | Coverage@10       |  NQ  | TriviaQA (w/o alias)  | TriviaQA (with alias) | WebQ | QAMPARI |
> |-------------------|:----:|:---------------------:|:---------------------:|:----:|:-------:|
> | DPR (multi)       | 67.9 |          67.3         |          17.9         | 58.8 |   16.9  |
> | Ours (sampling)   | 56.6 |          74.5         |          19.6         | 59.8 |   16.7  |
> | Ours (clustering) | 61.7 |          76.5         |          20.4         | 62.1 |   18.7  |
>
> **We updated Table 4 (Page 9) with the new Coverage@10 metric. We have not yet added the experimental results on Qampari to the main paper, which would require running all baseline methods and our GenRead under different settings. The table above shows the Coverage@10 on Qampari, which validates the coverage performance for multiple-answer scenarios.**
>
> **Please kindly let us know if the experiments on Qampari have addressed your concern! If so, we could add these experiments (after running all baselines) to the main paper. If not, could you please tell us which experiments should be added further?**
>
> [2] QAMPARI: : An Open-domain Question Answering Benchmark for Questions with Many Answers from Multiple Paragraphs. 2020. https://arxiv.org/abs/2205.12665

---

> > ### Comment · Reviewer_Vpy9 · 2022-12-05
> > **Thanks for adding the results on QAMPARI**
> >
> > Thank you for adding the result. It looks like even though there is an improvement in coverage, it's only somewhat minor (< 2 points). Also, do you know why the coverage of NQ is that much worse for your method? I think it's worth finding out and adding to the paper. Regarding adding the results of the QAMPARI dataset to the paper, I think it would be a good idea.

---

> > > ### Author Response · Authors · 2022-12-07
> > > **Response to results on QAMPARI**
> > >
> > > Thanks for your suggestion, we will add the results to the paper! It is worth mentioning that TriviaQA and QAMPARI have more than 13 possible answers, so the absolute value of coverage over all possible answers is relatively low. In fact, compared with the baseline, we have a relative 10% improvement.
> > >
> > > Besides, we do have a discussion "why our method has a lower coverage than DPR on NQ". The discussion is at Appendix A.7 (page 17). The reason is mainly due to the time-dependent and incomplete answer issues.

---

> ### Author Response · Authors · 2022-11-15
> **Response to Reviewer Vpy9 [1/4]**
>
> Thanks a lot for your reviews! Your professional reviews offer us great advice towards writing a more comprehensive and competitive paper! And, we are very encouraged that you found our proposed method simple and effective!
>
> > Q1. A weakness of the proposed model is that the clustering and selecting of the prompts is *independent* of the given question. Essentially, if the same n (question, document) pairs are selected from each cluster, the prompts are exactly the same for each question. This might be suboptimal. For example, do you have a sense of how many prompts on average are useful for retrieval for a given question? It would also be interesting to devise question-dependent prompting.
>
> **We agree with the suboptimal issue of using *question-independent* prompts. However, we tend to consider this to be orthogonal to the main goal of our paper. The reason is as follows.**
>
> First, the primary goal of our proposed clustering prompt is to select distinct in-context demonstration as prompts, in order to generate *multiple diverse documents* that cover different perspectives, leading to better recall/coverage over all possible answers. Extensive experiments show our method can achieve better recall/coverage score, compared to sampling methods under a single prompt.
>
> **Second, we believe a better in-context demonstration selection could be further inserted into our clustering prompt pipeline.** For example, we still do K-means clustering first, but instead of using the same in-context demonstration, we select question-dependent in-context demonstration for each question **under each cluster**. We noticed some recent works, e.g., [1], using reinforcement learning to select better in-context demonstration. **Adding the in-context demonstration selection is likely to bring extra improvement but might also make the pipeline more complex.**
>
> Again, we aim to generate diverse contextual documents and have proposed a simple clustering-based prompting method. We tend to think that selecting question-dependent in-context demonstration is a bit beyond the scope of our paper, wihch could be a further extension of our work in the future.

---

### Author Response · Authors · 2022-11-19
**Global Response: Summary of Paper Changes**

Dear Reviewers,

Thank you very much for reviewing our paper! Your professional reviews offer us great advice towards writing a more comprehensive and competitive paper! And, we are pleased that all reviewers appreciated the contributions of our work. To summarize our contribution, we present a novel perspective for solving knowledge-intensive tasks by replacing document retrievers with large language model generators. We demonstrate the effectiveness of generation under two reader variants (i.e., zero-shot and supervised settings), and establish new SoTA performance on multiple benchmark datasets.

As the phase 1 discussion approaching the end, we would like to summarize the changes to the paper in the updated version.

Newly added contents based on reviewer comments:

- [Page 9 Section 5] Limitation and future work discussion (reviewer Vpy9 and jnAy)
- [Page 17 A.6] Discussion on DPR and GPT inference cost (reviewer 2jcY)
- [Page 18 T12] Error analysis on retrieval and reading (reviewer jnAy)
- [Page 20 T15] Case studies on hallucination errors (reviewer Vpy9 and 2jcY)
- [Page 25 B.1] Prompt choices and corresponding performance (reviewer 2jcY)

Revised contents based on reviewer comments:

- [Page 1] Rewrite some sentences in the introduction and related work (reviewer Vpy9 and a9DF)
- [Page 9 Section 4.3.2] Coverage analysis using Coverage@K (reviewer Vpy9)
- [Page 10] Ethics statement on potential bias of language models (reviewer Vpy9 and a9DF)

We think these changes have improved the paper and again thank reviewers for suggesting them! If you still have any concerns, please kindly let us know and we will be happy to answer your further questions.


Best regards,

GenRead authors

---

### Author Response · Authors · 2022-11-28
**Global Response: Ethics Statement**

Dear Ethics Chair and Reviewers,

We have updated the ethics statement in the revision to discuss (and address at some extent) potential issues related to the research problem we study. We hope it helps practitioners and researchers become aware of the issues by the time they use the novel technical approach in our work.

Best regards,

GenRead authors

**==Ethics Statements==**

Large language models have a wide range of beneficial applications for society, but they also have potentially harmful applications. Previous work has shown various forms of bias, such as racial and gender bias, in large language models like GPT-3, even after explicit efforts to reduce toxic language [1]. The importance of addressing these societal harms is acknowledged by OpenAI themselves in their 2020 paper introducing GPT-3 [2], which stated “we focus on two primary issues: the potential for deliberate misuse of language models like GPT-3 ... and issues of bias, fairness, and representation within models like GPT-3.” on page 34.

The goal of this paper is to utilize knowledge stored in the parameters of large language models to solve knowledge-intensive tasks.
Unlike retrieve-then-read where an external corpus can be curated to be trustworthy, the use of a model to generate contextual documents may further permeate existing biases in common models. First, our work shows that generated documents suffer from challenges of stale information from outdated documents used for training.  Second, we show that generated documents tend to be less diverse, potentially biasing answers towards more common entities and terms from the training data. Finally, we conducted experiments on only three large language models. It is possible that some of our conclusions or observations may not necessarily hold for other models trained with different data or objectives.

Regarding ethical solutions, future work includes (i) further exploring potential bias and intentional or unintentional harm that may result from using generated contextual documents; (ii) better aligning language models with user intent to generate less biased contents and fewer fabricated facts.

[1] GPT-3 and InstructGPT: technological dystopianism, utopianism, and “contextual” perspectives in ai ethics and industry. AI and Ethics 2022.

[2] Language models are few-shot learners. Neurips 2020.

---

### Decision · Program_Chairs · 2023-01-20

**Decision:**

Accept: poster

**Justification For Why Not Higher Score:**

While the reviews of this paper are strong, it focuses on a fairly narrow task of knowledge intensive tasks. Further, it doesn't provide an in depth study of the exact phenomena that is observed (which is that why generate-and-read is better than retrieve-and-read). That being said, I wouldn't mind seeing this as a spotlight paper.

**Justification For Why Not Lower Score:**

All reviewers appreciated the technical contribution and the well execution of the ideas.

**Metareview: Summary, Strengths And Weaknesses:**

This paper proposes a "generate-then-read" approach for open-domain question answering that is based on large language models. It includes two models, one for the zero-shot setting and the other for the supervised setting. Experiments on 7 datasets show that the model outperforms several competitive baselines. Experiments on several datasets show the model outperforms a range of competitive baselines.

The reviewers liked the idea and found the work attractive, in the sense that it relies on the the internal knowledge stored in an LLM without a retrieval process from external sources. The strong empirical results and the wide range of experiments conducted were also appreciated in the reviews. Most concerns were addressed during the discussion. This paper opens up a new question of why this approach is better than retrieve-and-generate. There were suggestions of the need for further analysis or insights on this. Perhaps a more thorough investigation of this particular question may require a separate follow-up study.






**Note From Pc:**

if the above contains the word "oral" or "spotlight" please see: "oral" presentation means -> notable-top-5% and "spotlight" means -> notable-top-25%. As stated in our emails, we are disassociating presentation type from AC recommendations